# Mother brain is wired for social moments

Ortal Shimon-Raz[1,2†], Roy Salomon[3†], Miki Bloch[4,5], Gabi Aisenberg Romano[4,5], Yaara Yeshurun[6], Adi Ulmer Yaniv[1,3], Orna Zagoory-Sharon[1], Ruth Feldman[1]*

[1]IDC Herzliya, Bar Ilan University, Ramat Gan, Israel; [2]Department of Psychology, Bar Ilan University, Ramat Gan, Israel; [3]Gonda Brain Research Center, Bar Ilan University, Ramat Gan, Israel; [4]Department of Psychiatry, Tel Aviv Sourasky Medical Center, Tel Aviv, Israel; [5]Sackler Faculty of Medicine, Tel Aviv University, Tel Aviv, Israel; [6]School of Psychological Sciences, Tel Aviv University, Tel Aviv, Israel

**Abstract** Reorganization of the maternal brain upon childbirth triggers the species-typical maternal social behavior. These brief social moments carry profound effects on the infant's brain and likely have a distinct signature in the maternal brain. Utilizing a double-blind, within-subject oxytocin/placebo administration crossover design, mothers' brain was imaged twice using fMRI while observing three naturalistic maternal-infant contexts in the home ecology; 'unavailable', 'unresponsive', and 'social', when mothers engaged in synchronous peek-a-boo play. The social condition elicited greater neural response across the human caregiving network, including amygdala, VTA, hippocampus, insula, ACC, and temporal cortex. Oxytocin impacted neural response primarily to the social condition and attenuated differences between social and non-social stimuli. Greater temporal consistency emerged in the 'social' condition across the two imaging sessions, particularly in insula, amygdala, and TP. Findings describe how mother's brain varies by caregiving experiences and gives salience to moments of social synchrony that support infant development and brain maturation.

**\*For correspondence:**
feldman.ruth@gmail.com

†These authors contributed equally to this work

**Competing interests:** The authors declare that no competing interests exist.

## Introduction

Research into the brain basis of maternal care in mammals identified a set of subcortical limbic structures, which, primed by the oxytocin surge during labor, triggers the species-typical caregiving behaviors that usher young to social living (*Feldman, 2015a*; *Numan and Young, 2016*). These brief moments of social contact when mothers express the prototypical behavior of their species carry profound effect on infant sociality; reorganize the infant's lifetime oxytocin system (*Champagne et al., 2001*; *Feldman, 2016*; *Francis, 1999*; *Krol et al., 2019*), augment the salience of social cues (*Marlin et al., 2015*), and sculpt the infant's brain and behavior to life within the social ecology (*Hammock, 2015*). While few functions are as conserved as maternal care, in humans the subcortical structures that underpin mammalian mothering expanded to include insulo-cingulate, temporal, and frontal regions which coalesce to form the 'human caregiving network' (*Feldman, 2017*). Activation of this network supports the human-specific caregiving behavior and enables parents to perform the complex task of preparing human children to life within cultural communities; empathize with the infant's emotion, mentalize to infer infant intentions, prioritize caregiving activities, and plan for long-term parenting goals based on culturally-transmitted social values (*Feldman, 2015a*; *Feldman, 2017*).

Consolidation of the human caregiving network during the postpartum months is impacted by the mother's physiological and mental state and links with circulating oxytocin (OT) (*Atzil et al., 2011*) and cortisol (*Seth et al., 2016*) levels, degree of maternal anxiety and depression (*Pawluski et al., 2017*), and representations of own caregiving (*Kim et al., 2010*). Activation of the caregiving network also underpins the expression of mother-infant behavioral synchrony, the temporally matched repetitive-rhythmic social play which is observed universally and marked by

episodes of shared gaze, mutual positive affect, and 'motherese' high-pitched vocalizations (*Feldman, 2007*). Albeit brief, these precious social moments of synchrony expose mothers and infants to massive amounts of social inputs, require coordinated behavior to regulate the high positive arousal, and carry profound effects on infant sociality (*Tronick, 1989*). Longitudinal studies have shown that mother-infant synchrony plays an important role in children's socialization, emotion regulation, and stress management (*Feldman et al., 2010b*; *Feldman et al., 2013*). Furthermore, organization of the parent's caregiving network in infancy shapes children's social-emotional skills as mediated by behavioral synchrony and parental OT (*Abraham et al., 2018*; *Abraham et al., 2016*; *Kim et al., 2015*), highlighting the links between the three components of bonding; the caregiving network, OT system, and synchrony. Indeed, when bonding is disrupted, due to conditions such as postpartum depression or environmental stress, deficits are observed in all three components; activation of the parent's caregiving network, synchronous parenting, and OT production and these carry long-term effects on the child's propensity for psychopathology and maturation of neural systems that underpin empathy and attachment (*Davis et al., 2017*; *Kim et al., 2016*; *Levy et al., 2019*; *Pratt et al., 2019*).

Across species, the 'maternal care' envelope marks the overall provisions transmitted from one generation to the next that contain the evolutionary-acquired information necessary for survival and program the infant's brain to what it means to be a member of that species (*Kundakovic and Champagne, 2015*; *Meaney, 2001*). Maternal care comprises a range of long and arduous activities, such as nest building, food retrieval, and, in some primate species, group collaborative and defensive activities (*Hayes, 2000*; *Russell, 2003*). Episodes of maternal social contact interfacing with an individual infant are brief, and, in some species, last no longer than several minutes per day for several days (*González-Mariscal, 2007*; *Lucion and Bortolini, 2014*). In humans, moments of direct maternal-infant social contact are similarly brief and occupy a fraction of the overall maternal caregiving. Between three and nine months, the sensitive period for social development, episodes of mother-infant face-to-face synchrony typically last 3–5 min, but their impact is long-lasting (*Cohn and Tronick, 1988*; *Feldman, 2015b*). One mechanism that underpins the long-term effects of these brief social moments is *bio-behavioral synchrony* (*Feldman, 2017*). Moments of mother-infant behavioral synchrony provide a template for the coordination of physiological processes, allowing the mature brain to externally regulate the infant's brain and tune it to social living (*Hofer, 1994*; *Leong et al., 2017*). During synchronous play, mothers and infants coordinate their heart rhythms (*Feldman et al., 2011b*), oxytocin response (*Feldman et al., 2010a*), and neural oscillations (*Leong et al., 2019*), and these carry an 'imprinting-like' effect on the infant's brain. It is thus likely that these intense social moments also have a distinct signature in the maternal network.

In the current study, we examined whether mother-infant social moments marked by increased synchrony would trigger increased activations across the caregiving network in postpartum mothers. The human caregiving network comprises the inter-connected functioning of its subcortical (amygdala, VTA), para-limbic (AI, ACC), temporal (STS/STG, TP), and frontal (mPFC) components into a functional network that coalesces to support human caregiving (*Feldman, 2015a*; *Feldman et al., 2019*; *Kim et al., 2016*; *Swain et al., 2014*). While the brain of any adult exhibits responses in regions of this network to infant cues (*Kringelbach et al., 2008*; *Rilling and Mascaro, 2017*), synchronous social moments are expected to activate a coherent response across the network that is stronger and more unified as compared to similar maternal-infant cues that do not contain a social component. Although such comparison has not yet been tested, it is reasonable to assume that since maternal care is a time-consuming, metabolically-costly endeavor, bearing critical impact on species continuity, the mother's brain would not activate to it full capacity when resources are needed for other tasks but would cohere to its full expression to sustain these brief moments of sociality.

In addition, we examined whether activations of the human caregiving network to social moments would show greater sensitivity to oxytocin administration as compared to similar episodes of maternal-infant presence that do not contain a social component. OT is an important modulator of the brain's social functions (*Zink and Meyer-Lindenberg, 2012*) and supports reorganization of the mother's brain following childbirth (*Insel and Young, 2001*). OT plays a critical role in neural plasticity at the molecular and network assembly levels and such plasticity augments the salience and reward value of the infant to its mother (*Marlin et al., 2015*; *Oettl et al., 2016*; *Valtcheva and Froemke, 2019*), and both experimental and knockout studies demonstrate the causal role of OT in the initiation of maternal social behavior (*Higashida et al., 2010*; *Lopatina et al., 2012*). Human

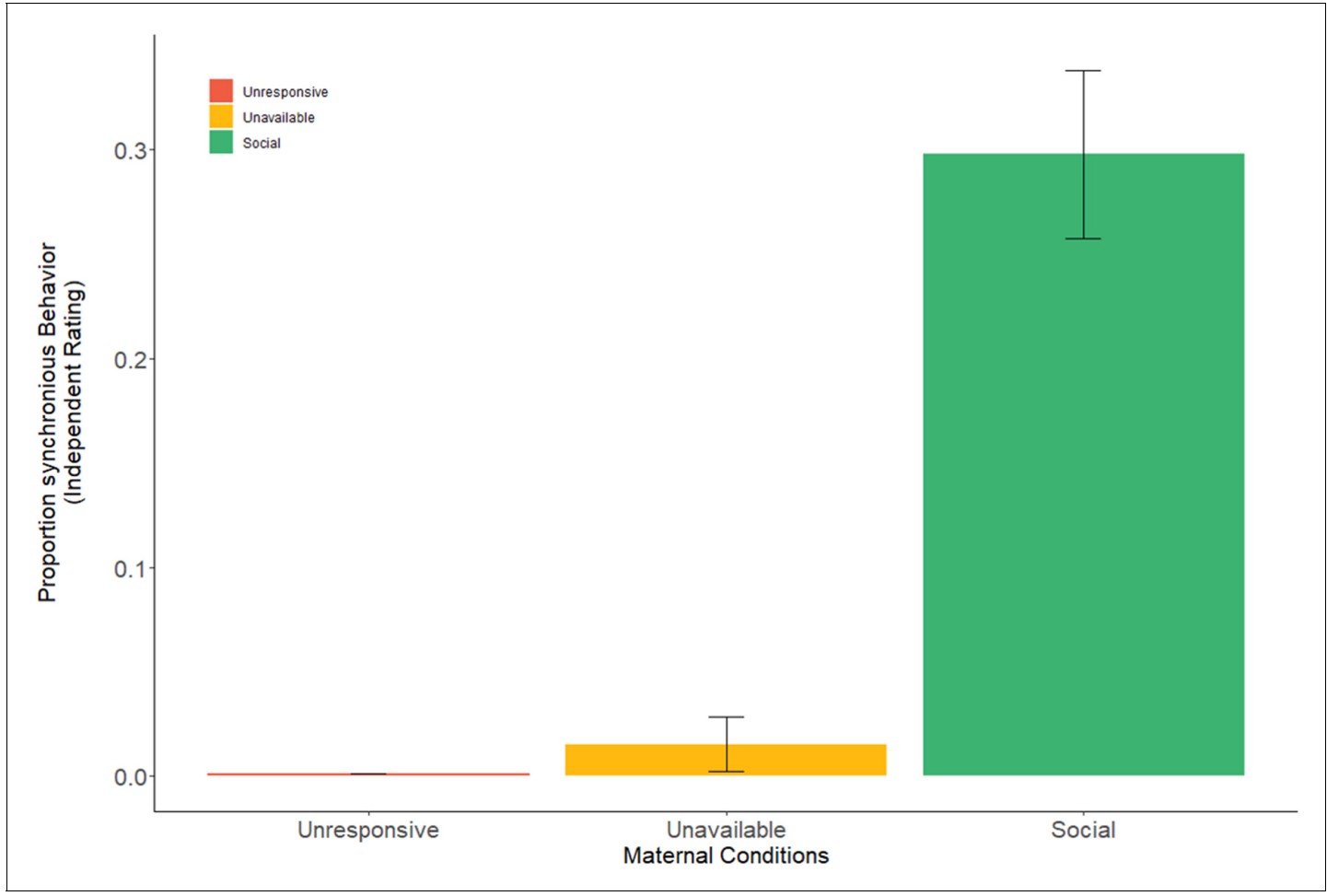

**Figure 1.** Proportion of mother-infant synchronous behavior in the three maternal conditions. Mother-infant synchrony occurred more during face-to-face interaction in the social maternal condition compared to the unavailable and unresponsive maternal conditions. All effects were Greenhouse-Geisser corrected. Error bars represent standard error of the mean.

The online version of this article includes the following source data and figure supplement(s) for figure 1:

**Figure supplement 1.** Mother salivary oxytocin levels (pg/mL) in the oxytocin and placebo conditions.

**Figure supplement 1—source data 1.** Results of analysis of effects within a two factors Bayesian repeated measures ANOVA (*PBO-OT×Time*).

studies have shown that peripheral OT levels are associated with mother-infant synchrony (*Feldman et al., 2011a*) and activation of the human caregiving network (*Abraham et al., 2014*; *Rilling and Mascaro, 2017*). Intranasal OT administration affects parenting by increasing parents' prototypical-rhythmic behaviors and augmenting parent-child synchrony (*Naber et al., 2010*; *Weisman et al., 2012a*) and OT may thus target the social context which is marked by high synchrony.

While we expected OT administration to affect primarily neural response to the social condition, the direction of its effects remained a research question. The effects of OT administration on BOLD response are far from clear and the literature is mixed on whether OT increases or decreases activations of nodes within the caregiving network (*Chen et al., 2017*; *Grace et al., 2018*; *Martins et al., 2020*; *Wang et al., 2017*; *Wigton et al., 2015*). Conceptually, whereas the 'social salience' hypothesis argues that OT increases the salience of social stimuli and to enhanced response to social signals (*Shamay-Tsoory and Abu-Akel, 2016*), the anxiolytic model on OT (*Neumann and Slattery, 2016*) may suggest that OT would level-out the increased response to social-emotional cues to maintain equilibrium and calm. Of the few studies that examined the effects of OT on the parent's neural response to infant stimuli, several indicated attenuation of BOLD response under OT. *Wittfoth-Schardt et al., 2012* tested fathers' neural response to unfamiliar, familiar, and own infant pictures

**Table 1.** Coordinates of activation peaks (whole brain ANOVA results).

Whole brain Coordinates are in MNI space. p<0.05 false discovery rate (FDR). L, left; R, right; BA, Brodmann's area; STG, superior temporal gyrus; TP, temporal pole.

| Anatomical area | BA | F (2,44) | p | Cluster size | Cluster peak voxel x | Y | Z |
|---|---|---|---|---|---|---|---|
| *'Maternal Condition' main effect* | | | | | | | |
| Cingulate gyrus | 31 | 8.47 | <0.001 | 389 | 12 | −22 | 40 |
| R TP-STG-insula | 41 | 94.49 | <0.00001 | 84971 | 54 | −25 | 4 |
| L TP-STG-insula | 41 | 88.30 | <0.00001 | 85828 | −54 | −22 | 4 |
| R superior frontal gyrus | 6 | 25.15 | <0.00001 | 8150 | 58 | -1 | 49 |
| L superior frontal gyrus | 6 | 23.99 | <0.00001 | 9609 | −27 | -7 | 52 |
| Bilateral supplementary motor cortex | 6 | 23.74 | <0.00001 | 10983 | 9 | 5 | 70 |
| R orbitofrontal | 11 | 9.82 | <0.0005 | 955 | 21 | 38 | −11 |
| L dorsolateral/prefrontal cortex | 9 | 8.31 | <0.001 | 318 | −33 | 41 | 34 |
| R occipital cortex | 18 | 12.24 | <0.0001 | 1304 | 12 | −79 | -5 |
| L occipital cortex | 18 | 8.27 | <0.001 | 572 | −21 | −94 | -5 |
| R cuneus | 19 | 16.18 | <0.00001 | 3038 | 24 | −82 | 40 |
| L fusiform | 37 | 8.96 | <0.001 | 689 | −36 | −64 | 0 |
| R parietal lobule | 5 | 21.85 | <0.00001 | 9305 | 27 | −43 | 58 |
| L parietal lobule | 5 | 18.17 | <0.00001 | 9723 | −30 | −40 | 49 |
| R basal ganglia- putamen | | 15.60 | <0.00001 | 4316 | 18 | 11 | 7 |
| L basal ganglia- putamen | | 17.28 | <0.00001 | 3046 | −21 | -4 | 10 |
| R paraippocampal gyrus | | 14.15 | = 0.00001 | 1575 | 36 | −40 | -2 |
| L parahippocampal gyrus | | 12.38 | <0.0001 | 2712 | −33 | −40 | −14 |
| R cerebellum | | 15.32 | <0.00001 | 1603 | 30 | −64 | −26 |
| L cerebellum | | 14.32 | = 0.00001 | 2686 | −30 | −61 | −26 |
| *Social>Unavailable* | | T (22) | | | | | |
| R TP-STG-insula | 41 | 12.07 | <0.00001 | 66205 | 63 | −22 | 4 |
| L STG-insula | 41 | 11.18 | <0.00001 | 69734 | −54 | −22 | 4 |
| R superior frontal gyrus | 6 | 7.41 | <0.00001 | 4134 | 60 | 5 | 40 |
| L superior frontal gyrus | 6 | 7.79 | <0.00001 | 6967 | −42 | 2 | 43 |
| Bilateral supplementary motor cortex | 6 | 5.64 | = 0.00001 | 7116 | -3 | -1 | 67 |
| R occipital cortex areas | 18 | 4.67 | = 0.0001 | 2281 | 12 | −79 | -5 |
| R parietal lobule | 40 | 5.16 | <.00005 | 4024 | 48 | −28 | 37 |
| L parietal lobule | 40 | 4.69 | = 0.0001 | 3031 | −42 | −40 | 49 |
| R basal ganglia- putamen | | 5.39 | <0.00005 | 4028 | 15 | 8 | 7 |
| L basal ganglia- putamen | | 5.51 | = 0.00001 | 3673 | −21 | -1 | 0 |
| R cerebellum | | 4.80 | <0.0001 | 1283 | 30 | −64 | −26 |
| L cerebellum | | 4.29 | <0.0005 | 1349 | −30 | −61 | −26 |
| *Social>Unresponsive* | | | | | | | |
| R TP-STG-insula | 41 | 10.36 | <0.00001 | 63180 | 60 | 2 | -2 |
| L TP-STG-insula | 41 | 9.65 | <0.00001 | 70134 | −54 | −22 | 4 |
| R superior frontal gyrus | 6 | 6.05 | <0.00001 | 7539 | 45 | 2 | 46 |
| L superior frontal gyrus | 6 | 6.40 | <0.00001 | 12392 | −42 | -1 | 55 |
| Bilateral supplementary motor cortex | 6 | 6.97 | <0.00001 | 13949 | -6 | 8 | 70 |
| L dorsolateral prefrontal cortex | 9 | 4.15 | <0.0005 | 495 | −33 | 41 | 31 |

*Table 1 continued on next page*

*Table 1 continued*

| Anatomical area | BA | F (2,44) | p | Cluster size | Cluster peak voxel | | |
|---|---|---|---|---|---|---|---|
| | | | | | x | Y | Z |
| R occipital cortex areas | 18 | 4.77 | <0.0005 | 2042 | −12 | −55 | 58 |
| R parietal lobule | 5 | 6.44 | <0.00001 | 8766 | 27 | −43 | 58 |
| L parietal lobule | 40 | 5.87 | <0.00001 | 10104 | −33 | −37 | 43 |
| R basal ganglia- putamen | | 4.39 | <0.0005 | 2136 | 21 | 2 | 10 |
| L basal ganglia- putamen | | 5.00 | = 0.00005 | 1539 | −21 | -4 | 13 |
| R cerebellum | | 4.31 | <0.0005 | 956 | 30 | −61 | −26 |
| L cerebellum | | 4.83 | <0.0001 | 1875 | −30 | −61 | −23 |
| *Unresponsive> Unavailable* | | | | | | | |
| R STG | 41 | 7.482 | <0.00001 | 3127 | 54 | −10 | 1 |
| L STG | 41 | 6.528 | <0.00001 | 413 | −45 | −19 | 1 |

and found increased response in subcortical reward regions, hippocampus, AI, STS, and OFC to own infant under placebo (PBO), which attenuated under OT, and concluded that OT attenuates neural response as a function of social salience. *Bos et al., 2018*, testing mothers, similarly showed increased response to own infant pictures under PBO, which decreased under OT in VTA, putamen, and amygdala and concluded that OT attenuates neural response as a function of social arousal. *Riem et al., 2016* showed amygdala attenuation under OT to infant cries pending maternal attachment representations. These studies lend support to the hypothesis that while the social condition would increase activations in the caregiving network under PBO, OT may level out these socially driven activations marked by salience and arousal. However, since other studies showed BOLD increases under OT in fathers' brain (*Li et al., 2017*) and as the current consensus is that OT effects are time-, person-, and context-sensitive (*Bartz et al., 2011*), we hypothesize that OT would target the social condition and explored the direction of its effects.

To describe mothers' neural responses to synchronous social moments, we expanded on a well-researched paradigm into the parental brain that utilized presentation of individually-tailored stimuli collected in the home ecology (*Atzil et al., 2011*; *Elmadih et al., 2016*; *Noriuchi et al., 2008*). We included three separate conditions that depict typical mother-child social and non-social contexts in the home environment. Across conditions, mothers were filmed sitting next to their child in the same level and distance, to control for differences in physical proximity and posture. In the first condition, mothers sat next to their infant while being otherwise engaged (Condition I, *Unavailable*); in the second, mothers sat facing the infant but did not engage in social interactions (Condition II, *Unresponsive*); in the third, mothers engaged in a prototypical, rhythmic social play of peek-a-boo (Condition III: *Social*). Mothers were imaged twice in a double-blind within-subject placebo-control design and observed the same three conditions viewing themselves (Self) and an unfamiliar mother-infant dyad (Other) once following administration of oxytocin (OT) and once after placebo (PBO).

We expected that the social condition would elicit greater response as compared to the other conditions across the caregiving network. Similarly, we hypothesized that OT would impact specifically the caregiving network's response to the social condition, and tested whether these OT-mediated responses would follow the 'social salience' hypothesis (i.e., increased brain activations to social condition under OT) or the anxiolytic model of OT (decreased brain activation under OT). In addition, and as an open research question, we explored whether activation of the caregiving network to the social condition would show greater consistency between the two imaging sessions as compared to the other conditions, particularly in the insula and structures of the temporal cortex (STS, TP). The social condition is characterized by repetitive-rhythmic social stimuli that may trigger distinct activations in the 'sociotemporal brain' (*Schirmer et al., 2016*), which gauges durations, patterns, and frequencies of temporally-ordered stimuli. We thus investigated whether the synchronous moments of mother-infant social play would elicit greater temporal consistency in limbic, insular, and temporal regions that underpin the brain's perception of temporal regularities. Finally, we expected that the mother's neural activations would show a differential response between her own and an

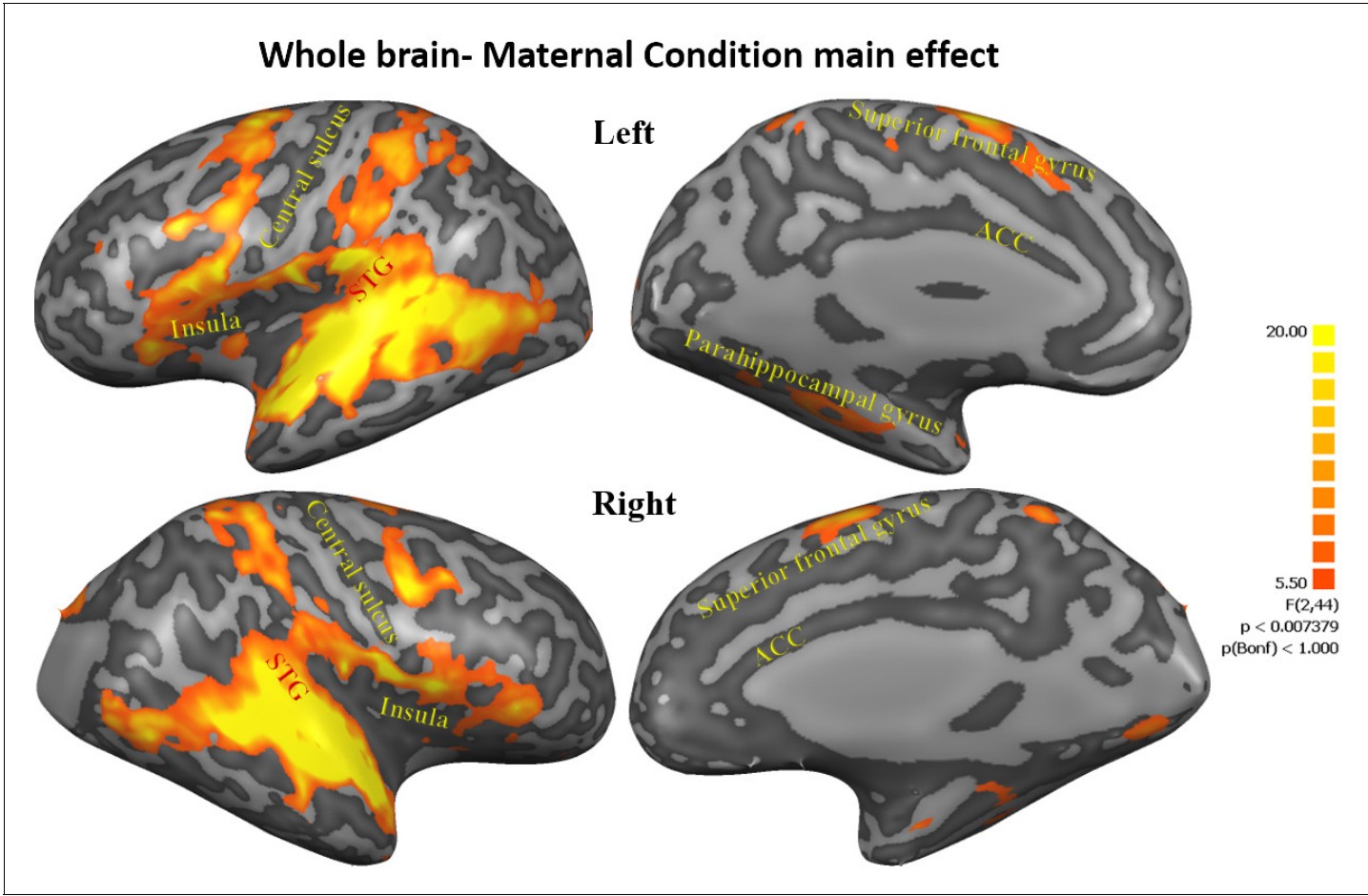

**Figure 2.** Maternal condition main effect. Figures representing activations from 3-factorial ANOVA 'maternal condition' main effect (FDR corrected, Cluster threshold 200 voxels) including the cingulate gyrus, bilateral insula, bilateral frontal lobe areas, bilateral STG to TP, bilateral parahippocampal gyrus, bilateral anterior cerebellum, bilateral basal ganglia- putamen, occipital lobe areas and right cuneus. STG, superior temporal gyrus; TP, temporal pole.

The online version of this article includes the following figure supplement(s) for figure 2:

**Figure supplement 1.** Map of Maternal Condition main effect.

unfamiliar infant and that these would increase in the social condition. To investigate if our results are specific to the caregiving network, we examined the effects of the social condition on two additional networks; the default mode network (DMN), a well-described network known to be activated by self-related processing (*Buckner et al., 2008*; *Peer et al., 2015*; *Salomon et al., 2014*; *Spreng et al., 2009*), and the visual network, an occipital task-positive network unrelated to bonding. We expected that while the DMN may show self-related effects, both the DMN and the visual neural systems would not be sensitive to the social versus non-social conditions or to OT administration, highlighting the specific response of the human caregiving network to social cues.

## Results

### Preliminary analysis: demonstrating high mother-infant synchrony during the 'social' condition and increase in salivary oxytocin levels following administration

To validate our procedure, we first examined whether synchrony levels (see Materials and methods for synchrony coding) were indeed higher in the social compared to the unavailable and unresponsive conditions, to ascertain that this condition exposed mothers to high levels of synchrony. As

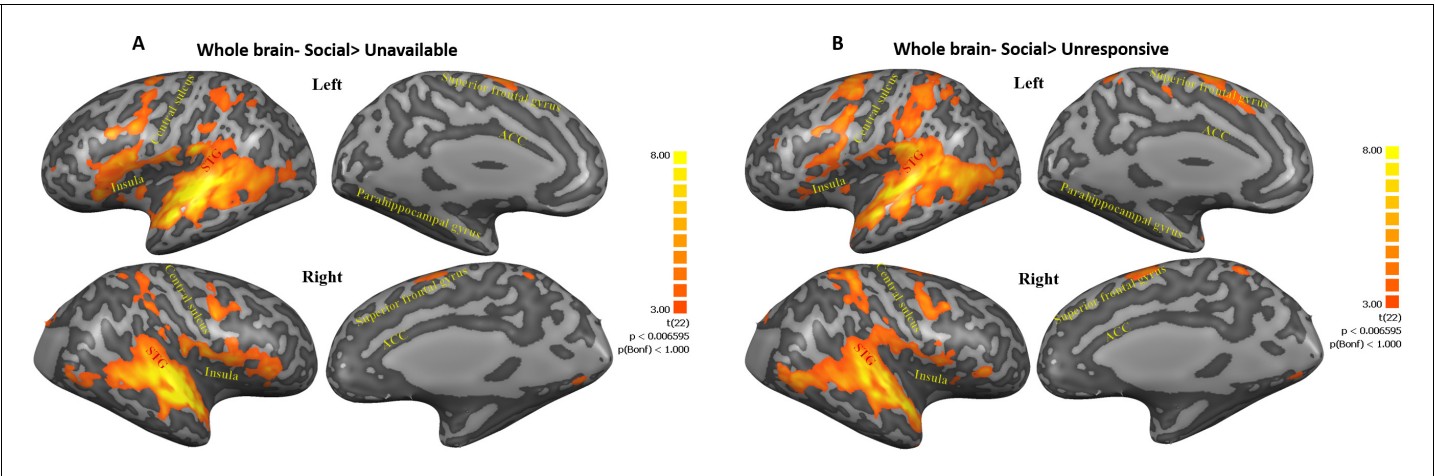

**Figure 3.** Maps of Social>Unavailable and Social>Unresponsive contrasts. (**A, B**) Figures represent regions within post hoc contrasts (Social>Unavailable; Social>Unresponsive, respectively) conducted to further examine the significant maternal condition main effect found in the whole brain three factorial ANOVA. Note that similar areas were elicited in both contrasts. This highlights the extensive activity along regions ranging from the insula, STG to TP and areas in the frontal cortex under the social condition. Subcortical structures in the basal ganglia and the cerebellum were also activated in both contrasts. Results are FDR corrected with Cluster threshold of 200. Brain regions are defined in *Table 1*. ACC, anterior cingulate; STG, superior temporal gyrus.

The online version of this article includes the following figure supplement(s) for figure 3:

**Figure supplement 1.** Map of Social>Unavailable contrast.

**Figure supplement 2.** Map of Social> Unresponsive contrast.

expected, a repeated measures ANOVA [F(1.16,25.56)=49.16, p<0.001] revealed that stimuli in the *Social* condition included significantly more synchrony (Mean = 0.298, SD = 0.194, 95% CI [0.219, 0.377]) compared to the *Unavailable* (Mean = 0.015, SD = 0.063, 95% CI [−0.011, 0.041]) and *Unresponsive* (Mean = 0.00, SD = 0.00) conditions (*Figure 1*), validating our paradigm. This effect for the social condition was supported by extremely strong evidence from a Bayesian repeated measures ANOVA conducted for social synchrony in the three maternal conditions ($BF_{incl}$ = 5.547e+11).

Next, to validate the OT manipulation, we tested whether peripherally-measured OT levels were indeed higher after OT administration. A 2 × 3 (*PBO-OT* × *Time*) repeated measures ANOVA on mother salivary oxytocin levels (pg/mL) showed a significant *PBO-OT* × *Time* interaction effect [F(1.1,24.09)=10.01, p<0.01], *PBO-OT* main effect [F(1,22)=10.46, p<0.01] and *Time* main effect [F(1.09,23.86)=11.03, p<0.01]. As expected, following OT administration mothers showed a marked increase in OT levels (Mean = 826.75, SD = 1135.25, 95% CI [362.795, 1290.705]) compared to the baseline (Mean = 21.49, SD = 13.61, 95% CI [15.928, 27.052]) and to the recovery samples (Mean = 193.86, SD = 303.56, 95% CI [69.801, 317.919]). Similarly, Bayesian analysis showed extreme evidence for *PBO-OT\*Time* interaction effect (BF = 197.08), as well as very strong evidence for *PBO-OT* (BF = 62.62) and *Time* (BF = 169.17) main effects (*Figure 1—figure supplement 1—source data 1*). In contrast, no significant increase in peripheral OT was observed following PBO administration (*Figure 1—figure supplement 1*).

## fMRI whole brain analysis

To examine brain regions associated with our conditions, a whole-brain three factorial ANOVA (*Maternal Condition* × *Self-Other* × *PBO-OT*) was calculated within BrainVoyager software. The analysis revealed a significant, FDR corrected, *Maternal Condition* main effect. A 200 voxels cluster size was used to extract volumes of interest (VOIs) from all regions that demonstrated significantly differential activity. The ANOVA revealed a widespread network of activations across the insula, superior-frontal and temporal areas in the cortex. Regions showing differential activations across the three maternal conditions included the cingulate gyrus, bilateral insula, bilateral frontal lobe areas, bilateral STG to TP, bilateral parahippocampal gyrus, bilateral anterior cerebellum, bilateral basal

**Table 2.** Results of 4 factors repeated measures ANOVA (ROI × Maternal Condition × Self-Other × PBO- OT) including seven preregistered ROIs defined as the maternal caregiving network.

|  | df | F score | P | Eta² |
|---|---|---|---|---|
| *ROI* main effect*** | 3.65, 80.24 | 20.63 | <0.001 | 0.48 |
| *Self-Other* main effect | 1, 22 | 0.54 | 0.471 | 0.02 |
| *Maternal Condition* main effect | 1.76, 38.63 | 2.35 | 0.12 | 0.1 |
| *PBO-OT* main effect | 1,22 | 0.01 | 0.945 | 0.00 |
| *ROI × PBO-OT* interaction | 3.19, 70.14 | 0.24 | 0.881 | 0.01 |
| *ROI × Maternal Condition* interaction * | 5.36, 117.85 | 2.7 | 0.021 | 0.11 |
| *ROI × Self-Other* interaction *** | 3.45, 75.87 | 18.73 | <0.001 | 0.46 |
| *PBO-OT × Maternal Condition* interaction** | 2, 43.89 | 6.92 | 0.002 | 0.24 |
| *PBO-OT × Self-Other* interaction | 1, 22 | 0.69 | 0.415 | 0.03 |
| *Maternal Condition × Self-Other* interaction | 1.67, 36.82 | 0.31 | 0.697 | 0.01 |
| *ROI × PBO-OT × Maternal Condition* interaction | 5.13, 112.93 | 0.95 | 0.452 | 0.04 |
| *ROI × PBO-OT × Self-Other* interaction | 2.79, 61.45 | 0.27 | 0.832 | 0.01 |
| *ROI × Maternal Condition × Self-Other* interaction | 5.71, 125.81 | 1.40 | 0.222 | 0.06 |
| *PBO-OT × Maternal Condition × Self-Other* interaction | 1.72, 37.94 | 0.4 | 0.644 | 0.02 |
| *ROI × PBO-OT × Maternal Condition × Self-Other* interaction | 5.53, 121.74 | 0.63 | 0.697 | 0.03 |

Bayesian analysis of ROI effects. Four factor Bayesian repeated measures ANOVA (*ROI × Maternal Condition × Self- Other × PBO- OT*).

In the table are results of 4 factors repeated measures ANOVA (*ROI× Maternal Condition× Self-Other× PBO-OT*). All results are Greenhouse-Geisser corrected. OT, oxytocin; PBO, placebo; *, p<0.05; **, p<0.01; ***, p<0.001.

The online version of this article includes the following source data for Table 2:

Source data 1. Results of analysis of effects within a 4 factors Bayesian repeated measures (ANOVA ROI× Maternal Condition× Self-Other× PBO-OT). BFincl is calculated using the Baws factor approach across all matched models.

ganglia-putamen, occipital lobe areas and right cuneus (see *Table 1*, *Figure 2*, *Figure 2—figure supplement 1*).

To examine the origin of the maternal condition main effect, three planned contrasts were used: *Social* (Self+Other+OT+PBO) > *Unavailable* (Self+Other+ OT+PBO) (*Figure 3A*, *Figure 3—figure supplement 1*) and *Social* (Self+Other+OT+PBO)> *Unresponsive* (Self+Other+OT+PBO) (*Figure 3B*, *Figure 3—figure supplement 2*). As seen in *Figure 3*, both contrasts elicited activations in the temporal and frontal cortices including the STG to TP, the insula, and the superior frontal gyrus, in addition to activations in subcortical structures in the basal ganglia (the putamen and the globus pallidus) and in the cerebellum, which were significantly higher in the *Social* compared to the other conditions, supporting our first hypothesis. The contrast of *Unresponsive* (Self+Other+OT+PBO) > *Unavailable* (Self+Other+ OT+PBO) was examined as well.

No significant, FDR corrected, results were found for *Self-Other* or *PBO-OT* main effects. All activations for all contrasts can be seen in *Table 1*.

## Oxytocin effects on ROIs activation

In order to examine the seven preregistered regions of interest within the maternal caregiving network, a factorial repeated measures ANOVA (*ROI × Maternal Condition × Self-Other × PBO-OT*) was performed on the beta values extracted from each of the ROIs. A significant main effect of *ROI* was found (*Table 2*). Importantly, no main effect of *PBO-OT* was found, indicating that the network was not significantly globally modified by OT administration. Bayesian analysis indicated strong evidence against system level effects of OT (BF = 0.052). Similarly, no significant main effect of *Self-Other* was found with moderate (BF = 0.131) evidence for an absence of an effect (*Table 2—source data 1*). No significant effect of *Maternal Condition* was found. Similarly, no *ROI× PBO-OT* interaction effect was found. No four-way interaction was found. All analysis results are presented in *Table 2*.

Contrary to our hypothesis, we did not find a *Self-Other* main effect. A significant *Self-Other × ROI* interaction effect was found (*Table 2*), indicating differential responsivity to *Self-Other* conditions across the network. Indeed, almost all (6/7) of the maternal caregiving network ROIs showed a differential response to self vs. other- stimuli. Sensitivity to Self-Other distinction was also found in the DMN (See Appendix 1; *Appendix 1—figures 1*, *2*; *Appendix 1—tables 1*, *2* for full self-other analysis details).

Interaction effect for *Maternal condition × ROI* was found (*Table 2*). Post hoc repeated measures ANOVA conducted in each of the ROIs (*Table 3*) revealed significant maternal condition main effect in the insula that is driven by high responses to social (Mean = 0.05, SD = 0.19, 95% CI [−0.028, 0.128]) compared to unavailable (Mean = −0.05, SD = 0.18, 95% CI [−0.123, 0.024]) and to unresponsive conditions (Mean = −0.08, SD = 0.17). In the parahippocampal gyrus this main effect was driven by a high response to unavailable (Mean = 0.10, SD = 0.15, 95% CI [0.039, 0.161]) and unresponsive (Mean = −0.01, SD = 0.16, 95% CI [−0.055, 0.075]) compared to social (Mean = −0.04, SD = 0.22, 95% CI [−0.129, 0.049]).

Critically, our main finding was defined by an interaction effect of *Maternal Condition × PBO-OT* in the maternal caregiving network (*Table 2*, *Figure 4A*). This was validated with a Bayesian Analysis indicating that the addition of ROI to the model (*ROI × Maternal Condition × PBO-OT*) had a BF = 0.0009 indicating very strong evidence for a lack of interaction with ROI (*Table 2—source data 1*). Post hoc tests revealed significant attenuation of brain response to the social condition after OT administration (t = 2.28, p=0.033), while such differences were not found in brain response to the *Unavailable* (t = −1.48, p>0.05) or to the *Unresponsive* conditions (t = −0.48, p>0.05). *Maternal Condition× PBO-OT* interactions for each of the maternal caregiving network ROIs are presented in *Figure 4B–F* for demonstration purposes. Parallel analyses of 3 factors repeated measures ANOVA *Maternal Condition × Self-Other × PBO- OT* was performed on both the DMN and visual system used to test the specificity of this response to the maternal caregiving network. In both the DMN and the visual system, no significant effect of *Maternal Condition × PBO-OT* interaction was found (see Appendix 1, *Appendix 1—figure 3*, and *Appendix 1—tables 3*, *4*). This reinforces the evidence for the specificity of the OT and social condition interaction in the maternal caregiving network.

## Within subject correlation (WSC)

Finally, we wished to explore the temporal consistency of activation patterns in the maternal network ROIs across the two scans for the three maternal conditions. For this, we used a Within Subject Correlation (WSC) approach for each participant between the oxytocin and placebo scans for each of the three maternal conditions. A three factorial repeated measures ANOVA (*ROI × Maternal Condition × Self-Other*) revealed a significant main effect of *ROI* [F (4.35, 95.59)=3.85, p=0.005], and *Maternal Condition* [F (1.96, 43.18)=5.33, p=0.009]. Post hoc comparisons revealed significantly stronger WSC under the *Social* condition (Mean = 0.13, SD = 0.11, 95% CI [0.085, 0.175]) compared to the *Unresponsive* condition (Mean = 0.02, SD = 0.10, 95% CI [−0.021, 0.061]) (t = 3.07, $p_{bonf}$ = 0.017) (*Figure 5A,B*, *Figure 5—figure supplement 1*). No other main effects or interactions were significant. Furthermore, we computed a Bayesian repeated measures ANOVA (*ROI × Maternal Condition × Self-Other*) and found strong evidence against inclusion of the Self-Other factor

**Table 3.** Results of significant ROI × Maternal Condition interaction effect within a repeated measures ANOVA (ROI × Maternal Condition × Self-Other × PBO- OT) separately for each of the preregistered ROIs.

| | Maternal caregiving network | | | | | | |
|---|---|---|---|---|---|---|---|
| | Insula | ACC | TP | Amygdala | VTA | PHG | NAcc |
| *Maternal Condition main effect, df (2,44)* | | | | | | | |
| F score | 6.31 | 1.82 | 2.63 | 0.99 | 1.92 | 7.32 | 1.42 |
| P | 0.006ᵚ | 0.176 | 0.089 | 0.376 | 0.167 | 0.002ᵚ | 0.25 |
| Eta² | 0.22 | 0.08 | 0.11 | 0.04 | 0.00 | 0.25 | 0.06 |

In the table post hoc analysis of significant interaction done separately for each of the preregistered ROIs. All results are Greenhouse-Geisser corrected. OT, oxytocin; PBO, placebo; ACC, anterior cingulate cortex; NAcc, nucleus accumbens; PHG, parahippocampal gyrus; TP, temporal pole; VTA, ventral tegmental area. *, p<.05; ψ, Bonferroni correction for multiple comparisons.

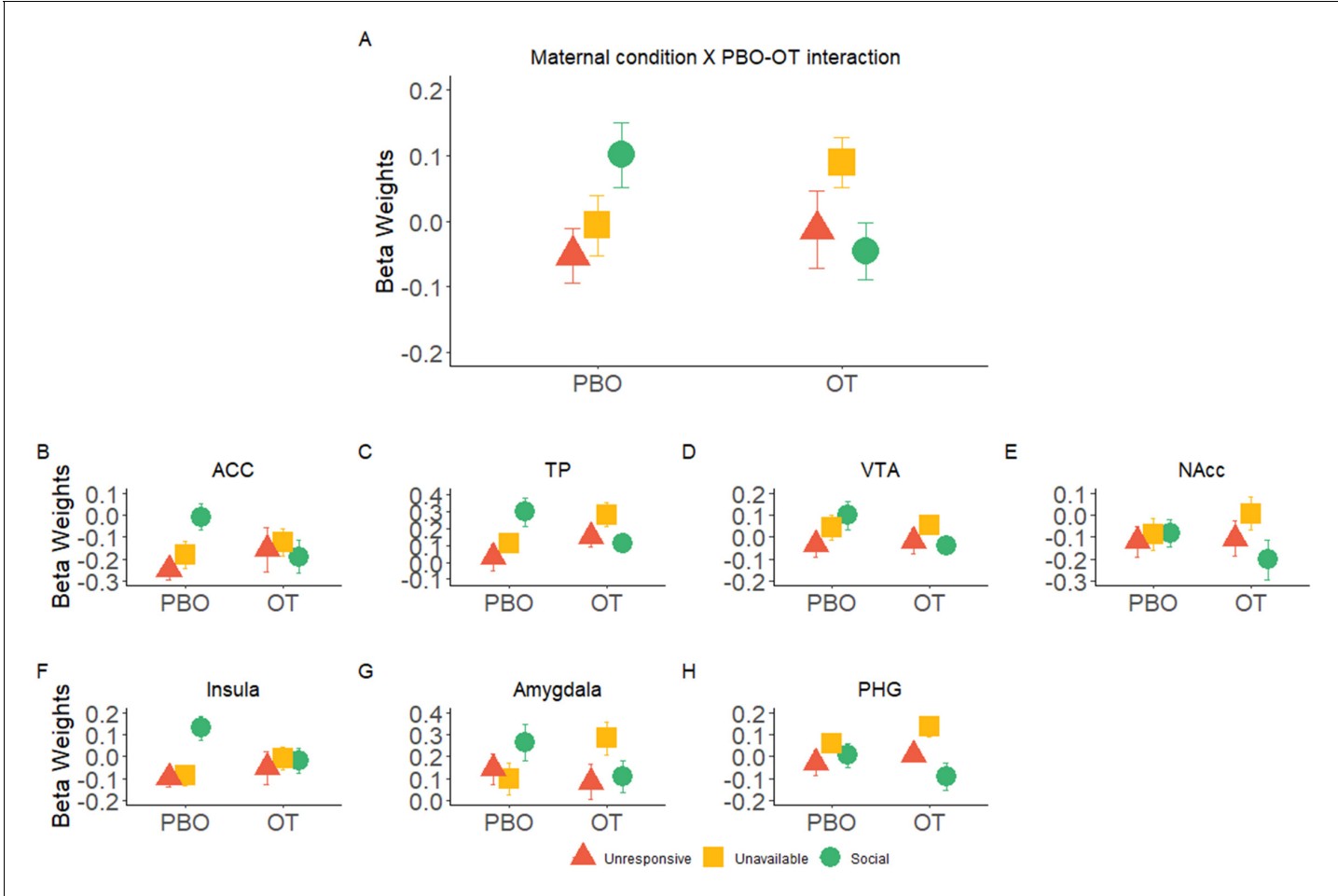

**Figure 4.** Significant interaction effects of *Maternal Condition × PBO-OT*. (**A**) A significant interaction effect of *Maternal Condition× PBO-OT* was found for the maternal network. This was driven by attenuation in brain response to the *social* condition after oxytocin administration. Brain response to the *Unavailable* and *Unresponsive* Maternal conditions did not differ between the two scans. (**B-F**) Interaction effects of *Maternal Condition × PBO-OT* in seven preregistered ROIs of the maternal brain shown for demonstration purposes only. Bars depict Standard error of the mean. OT, Oxytocin; PBO, placebo; PHG, parahippocmpal gyrus; ACC, Anterior cingulate; TP, Temporal pole; VTA, ventral tegmental area.

(BF = 0.005) (*Figure 5—source data 1*). Since no significant self-other main effects, nor self-other × maternal condition interactions were found, we averaged the 'self' and 'other' variables within each condition and ROI. In the DMN, no significant effect of maternal condition or Self-Other or interaction between them was found (see Appendix 1, *Appendix 1—figure 4*.).

Finally, an exploratory, post-hoc analysis was conducted to test brain-behavior coupling between mother-infant synchrony and child social engagement measured during a separate observation in the home ecology and neural activations in the social condition. We specifically looked for correlations with the 'social' condition that mirrors the mother-infant social interaction patterns in their natural habitat. Pearson's correlations showed significant associations between activations in the VTA, ACC, and insula and mother-child behavioral patterns (see Appendix 1, *Appendix 1—figure 5*, *Appendix 1—table 5*). However, these findings should be treated with caution due to the low number of participants.

## Discussion

For social mammals born with an immature brain that requires the external regulation of the mother for growth and development, moments of maternal-infant social interactions hold a special significance. These brief social moments integrate multiple well-orchestrated bio-behavioral processes

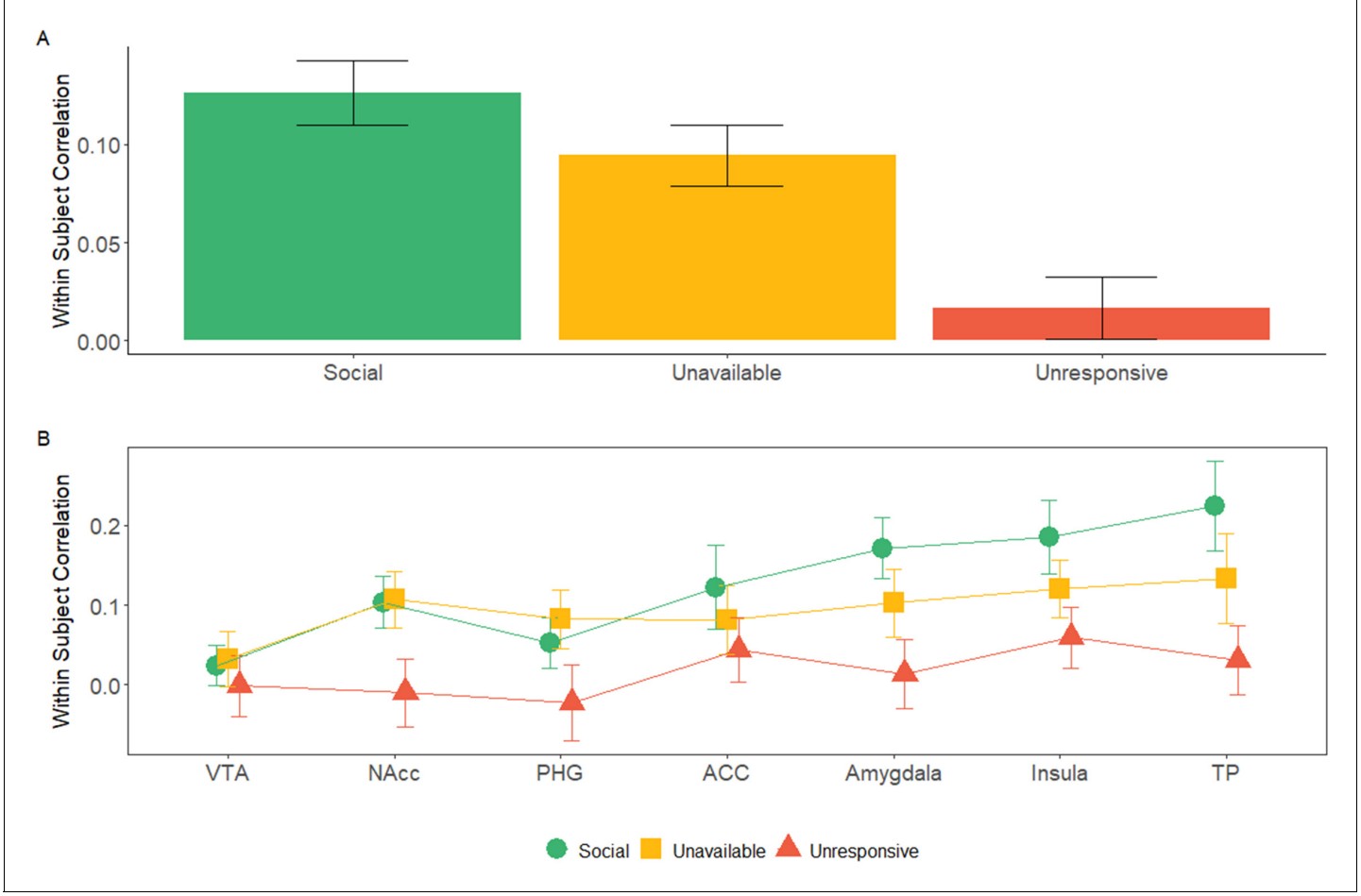

**Figure 5.** Within subject correlation (WSC) across maternal caregiving network ROIs: WSC represent the temporal activation pattern consistency of the ROIs under the three maternal conditions. It was calculated for each subject by Pearson's correlation for the BOLD time course of each condition between the OT and PBO runs. (A) 7 × 3 repeated measures ANOVA (*maternal caregiving network ROIs × Maternal Condition*) revealed a significant main effect of *ROI* and *Maternal Condition* but no interaction. Bayesian analysis indicated strong evidence for the absence of *ROI × Condition* interaction. Under the *social* maternal condition, the WSC was higher compared to the *unresponsive* maternal condition. (B) WSC under the social, unavailable and unresponsive conditions for each of the ROIS are presented for demonstration purposes only. Bars depict Standard error of the mean. ACC, Anterior cingulate; NAcc, Nucleus accumbens; PHG, parahippocmpal gyrus; TP, temporal pole; VTA, Ventral tegmental area. comparisons between ISC under the social and unresponsive conditions for each of the ROIS are presented for demonstration purposes only. Bars depict Standard error of the mean. OT, oxytocin; PBO, placebo; ACC, Anterior cingulate; NAcc, Nucleus accumbens; PHG, parahippocmpal gyrus; TP, temporal pole; VTA, Ventral tegmental area.

The online version of this article includes the following source data and figure supplement(s) for figure 5:

**Source data 1.** Results of analysis of effects within a three factors Bayesian repeated measures ANOVA (*ROI× Maternal Condition× Self-Other*).
**Figure supplement 1.** Graphs depicting the WSC for Social and Unresponsive conditions across all ROIs of the maternal caregiving network.

that consolidate the caregiving network in the maternal brain, trigger the species-typical caregiving behavior, and carry long-term impact on the developing infant brain (*Feldman, 2020*). Our study uniquely tests the response of the human caregiving network in the maternal brain to these social moments, versus other moments of non-social mother-infant presence, to shed further light on how human mothers' brains may change following birth and vary with the caregiving experience with one's infant.

To tap the response of the caregiving network to social interactions, we employed a double-blind, within subject, OT/placebo crossover design; hence, one of the novel aspects of our study is the closely-scheduled repeated imaging of the mother's brain in response to multiple ecological contexts during the sensitive period of bond formation. Findings describe several processes by which social moments impact the maternal network. First, whole-brain analysis (*Figures 2* and

3) demonstrated a significant widespread response to the social condition across large expanses of the brain, including the human caregiving network. This substantial and integrated response across temporal, frontal, and insular cortices, as well as subcortical regions, was specific to the social context, which elicited much larger neural activations in comparison with both the 'unavailable' and 'unresponsive' conditions. While these conditions included familiar mother-infant daily stimuli, a similar degree of physical proximity, and the same mother-infant sitting posture, they did not include the synchronous social component (see *Figure 1*). Moreover, the specific activations to the social condition was not found for the other systems we tested: the DMN, which underpins self-related processes, and the low level visual system regions, suggesting that our findings do not merely reflect a widespread neural response but are specific to the human caregiving network. These findings support our first hypothesis that these brief and universal synchronous social moments uniquely elicit a widespread response in the caregiving network, which sustain human maternal care, and that such differentiation of social from non-social mother-child episodes is specific to this network.

Second, we show that the effects of OT on the caregiving network were more notable during the social condition. OT decreased brain activation to the social condition throughout the caregiving network, indicating that the mother's neural response to synchronous social moments is more sensitive to OT. Bayesian analysis provided support for a coherent network-wide response to the social condition under OT across the caregiving network regions of interest (see *Figure 4*), further supporting the proposition that the interaction of social cues and OT impacts the network-wide functioning of this network. The special sensitivity of these socially driven neural activations to OT administration showed a consistent pattern which was similar across the pre-registered ROIs comprising the caregiving network (see *Figure 4B*). While these regions increased activity to the social context under natural conditions (i.e. PBO), OT leveled out these socially driven activations and under OT no significant differences were found between the social and non-social conditions.

To date, studies on the effects of OT administration on BOLD response have yielded mixed results and no straightforward explanation. Still, our findings are consistent with the majority of prior research which showed that OT targets social functions. OT enhances social behaviors, increases social collaboration, and augments understanding of social cues, and OT's effects on the brain are sensitive to social salience (*Di Simplicio et al., 2009*; *Guastella et al., 2008*; *Hurlemann et al., 2010*). Our findings are also consistent with research on OT's effects on the parental brain, which showed increased activation to own-infant pictures under PBO across wide areas of the caregiving network that were attenuated under OT, and the authors considered these results to stem from the social salience and high arousal embedded in these stimuli (*Bos et al., 2018*; *Wittfoth-Schardt et al., 2012*). A recent study testing the effects of OT administration via both intravenous and intranasal pathways on neural response during resting state showed that OT via both pathways decreased activity in amygdala, insula, TP, and parahippocampal gyrus, and the decrease was mediated by elevations in peripheral OT levels (*Martins et al., 2020*). Our findings similarly show a combination of increase in peripheral OT levels following OT administration and attenuation of neural response to social cues in the same regions. Our results may extend the resting state findings to include the caregiving network's processing of ongoing live social stimuli. In combination, these results may suggest that one pathway for the anxiolytic effects of OT may relate to the attenuation of activity in a limbic network that monitors salience cues, gauges danger signals, and integrates exteroceptive and interoceptive inputs to give immediacy and focus to ongoing events.

While our findings may be consistent with some prior research, it is important to note that OT's effects on BOLD response is complex and there is probably no simple relationship between social significance and consistent increase or decrease in levels of activations. To date, the effects of OT administration on the human brain have mainly been tested via alterations in BOLD response; however, much further research is needed along several additional dimensions and the application of more finely-tuned analytic tools. OT effects may, for instance, express in augmenting the brain's sensitivity to temporal regularities during the processing of social stimuli and further research should examine the conditions, specific patterns, and populations under which such sensitivity is enhanced or suppressed.

One possible cause for the associations between OT and the processing of temporal patterns in the mother's brain may relate to the crosstalk between OT and dopamine (DA) that underpins bonding. The initiation of maternal care in rodents involves a two-step process; first, OT leads to long-term depression in amygdala to suppress social avoidance of infant stimuli (*Gur et al., 2014*), and

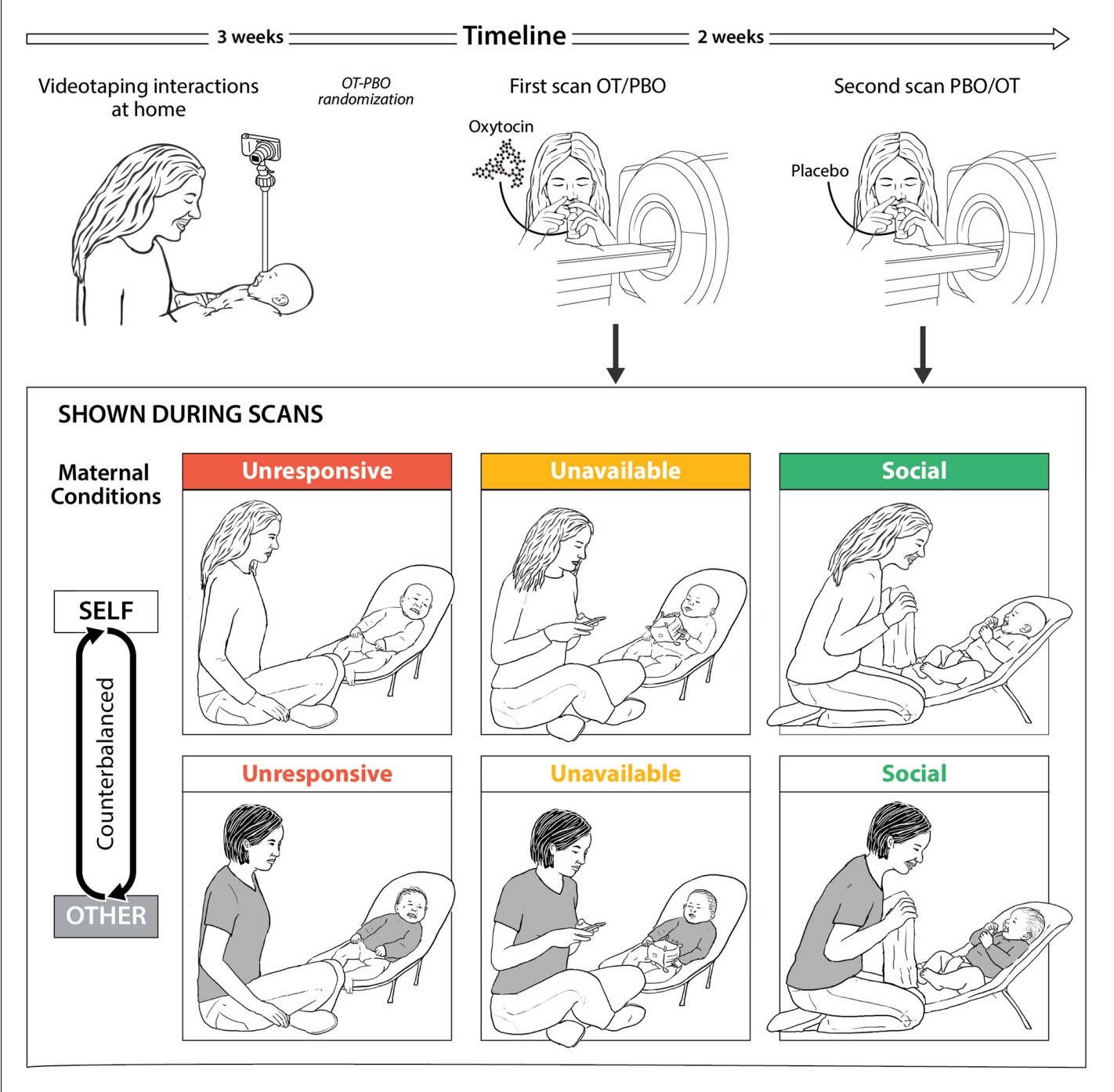

**Figure 6.** Research plan and fMRI paradigm. (**A**) Experimental procedure. Mothers and infants were recruited 4-7 months post-partum and videotaped during a home-visit in the first session. Video vignettes of interactions were used as fMRI stimuli. During the second and third sessions mothers administered oxytocin or placebo before participating in brain scanning, in a randomized, placebo-controlled, double-blind, two-period crossover design. On average two weeks elapsed between scans. (**B**) Experimental paradigm. Mothers were presented with six video vignettes of self and other (fixed control stimuli) mother-infant interactions depicting three maternal conditions: Unresponsive- no interaction with the baby, mother does not respond to the baby (shown in red), Unavailable- minimal interaction, mother is busy, but respond to the baby when he/she signals (shown in yellow) and Social-mother and infant are engage in a peek-a-boo face-to-face social interaction (shown in green). Clips lasted 1 min each and were previewed by rest with fixation period of 1 min. A rest with fixation periods of alternately 15-18 seconds was presented between clips. Order of self-other and maternal conditions were counterbalanced between the two scans (PBO/OT).

next, OT connects with DA through striatal neurons that encode for both OT and dopamine D1 receptors (*Olazábal and Young, 2006*). This enables DA neurons that encode sensory-motor general reward patterns to also encode the temporal patterns of *social* reward (*Báez-Mendoza and Schultz, 2013*; *Ross and Young, 2009*). This process allows the brain to internalize the social partner and its preferences, encode relationship-specific socio-temporal patterns, and draw reward from the matching of the social actions of self and partner, that is, social synchrony (*Báez-Mendoza and Schultz, 2013*; *Schultz, 2016*). The crosstalk of OT and DA triggered by maternal care enables the mother's brain to integrate rewarding experiences from the infant's smell, touch, babbling, and cute face into an overall representation that contains the dyad-specific temporal rhythms to sustain the attachment bond (*Ross et al., 2009*).

In addition to reward, OT also has well-known anxiolytic properties (*Neumann, 2008*) and in the context of maternal care the soothing function of OT may be important for survival. During labor, OT surges to levels much higher than the body's daily levels and these function to sooth the mother's pain and stress through the regulatory effects of OT on the hypothalamic-pituitary-adrenal-axis and sympathetic arousal (*Carter, 2014*; *Neumann and Slattery, 2016*). OT enables tranquility during the birth process, attenuates amygdala response to external events, and diminishes insular monitoring of unfamiliar interoceptive signals, keeping mother's brain from oscillating between extreme emotional states. Since OT administration leads to unusually high peripheral OT levels (*Weisman et al., 2012b*), the anxiolytic properties of OT during the postpartum period, particularly in response to infant stimuli, may have functioned here in a similar way to level-out the mother's neural response to different emotional states. However, this hypothesis is preliminary and requires much further research, possibly by using stimuli that target mothering-related stress and anxiety.

Our exploratory WSC analysis showed greater consistency in patterns of activation across the two imaging sessions in the social condition. Such socially driven consistency was found across the entire maternal caregiving network and a Bayesian analysis supported a network-wide temporal consistency across ROIs, particularly in amygdala, insula, and TP. It is important to emphasize that the WSC analyses were exploratory and should be interpreted cautiously. A repeated and highly arousing stimulus may in of itself elicit greater consistency across imaging sessions as compared to the other conditions that were more boring, less salient, and may generally entrain neural processes to a lesser degree. Similarly, the temporal consistency did not show differences related to own versus unfamiliar interaction and we did not tease apart conditions of infant alone, mother alone, and maternal-infant presence. Hence, these findings should be considered as very preliminary and future research should carefully tease apart the components related to salience from those related to self and those related to the mother's own attachment to her infant. Despite their very preliminary nature, the WSC analysis may point several important directions for future research on the neural basis of maternal-infant attachment and the effects of OT administration on the brain more generally. For instance, it appears that while the magnitude of BOLD responses in the caregiving network to social stimuli decreased under OT, the WSC analysis, which encodes the temporal pattern of response, was preserved. This suggests that regional BOLD fluctuations alone may not fully capture the neural processes induced by OT administration during social moments, and future studies may apply more novel multivariate spatial and temporal patterns analyses (*Saarimäki et al., 2016*; *Ulmer Yaniv et al., 2020*). Possibly, OT has a complex, multi-dimensional effect on the neural basis of social processes that impacts both the signal magnitude and the encoding of temporal patterns during the processing of social cues and both dimensions should be considered in future studies.

Another interesting outcome of the WSC analysis is pinpointing the specific regions that exhibited the greatest consistency in response to social stimuli; the amygdala, insula, and TP, nodes that have been implicated in the perception of temporal regularities, particularly in social contexts. For instance, amygdala neurons in the primate brain were found to simulate and foreshadow the partner's decision making process (*Grabenhorst et al., 2019*). Similarly, the insula and regions of the temporal cortex, including the STS and TP, monitor the salience and valance of stimuli and underpin the perception of temporal regularities and the duration and patterning of social stimuli (*Schirmer et al., 2016*). The insula plays a key role in interoception, the process by which the brain uses repeated experiences of own bodily signals to build predictions of self and others' physical and mental states (*Barrett and Simmons, 2015*; *Salomon et al., 2016*; *Seth, 2013*). Insular activations in the maternal brain enable mothers to represent the infant's bodily signals of hunger, fatigue, satiety,

and pain in their own brain (*Abraham et al., 2019*). The insula also serves as a center of allostasis, the brain resource-regulator function that sustains the mother's ability to detect patterned regularities in the infant's physiological needs and satisfy them in a timely manner before they arise (*Schulkin and Sterling, 2019*). Much further research is needed to explore the role of temporal regularities in the mothers' limbic, insular, and temporal response to social rhythms and their long-term effects on the infant's developing brain, social behavior, and the formation of the parent-infant attachment.

Although preliminary, we found evidence for brain-behavior coupling between patterns of mother-infant synchrony observed in the home environment and maternal neural response to her infant in the social condition in ACC, insula, and VTA, but not in other conditions. While these findings should be treated with great caution, due to the small sample size, they again show the selectivity of the social condition in linking maternal neural response to the degree of synchrony the infant experiences during daily playful moments in the home environment. Since mother-infant synchrony is an important dyadic experience that is both individually stable from infancy to adulthood and predicts social-emotional competencies, our preliminary findings suggest that this important dyad-specific pattern of synchrony may also be linked with activation of the mother's caregiving network. Such brain-behavior link between behavioral synchrony and mother's neural response to own infant in the social condition appears to be specific to the synchrony dimension of mother-infant interaction. The degree of infant social engagement in the interaction, while showing correlations with maternal VTA and TP response in the social condition, these correlations were not specific to own infant but emerged for both own and unfamiliar social interaction. Notably, the infant's temperamental dimensions of reactivity and emotionality were unrelated to maternal neural response in any condition, corroborating developmental perspectives which emphasize that attachment and temperament are two distinct processes (*Bowlby, 1969*; *Sroufe, 1985*). However, since we did not have standard measures to assess temperament, this should be considered a study limitation and our results underscore the need for much further research on the bi-directional associations between mother's brain, mother-infant interaction, and maternal and child factors.

Results indicate that nearly all regions of the caregiving network showed sensitivity to one's own vs. unfamiliar interactions. Contrary to our hypothesis, the directionality of the BOLD response varied across ROIs; higher activation for the 'Self' condition in the ACC and insular regions, and higher activity to the 'Other' condition in the other ROIs, with no differences in the VTA. These findings suggest that the caregiving network is highly responsive to the Self-Other distinction and exhibits a differential response to self-related mother-infant stimuli across six of seven ROIs. The ACC and insular regions respond to self-related stimuli across different types of stimuli (*Karnath and Baier, 2010*; *Northoff et al., 2006*; *Qin et al., 2012*; *Salomon et al., 2018*). Yet, in contrast to our expectation, no interaction effect was found between maternal condition and own-unfamiliar infant, indicating that the mother's neural response to social moments did not differentiate the two. Possibly, the salience of these arousing species-typical social exchanges is especially high and may trump differentiation of self and other, but further research is needed to understand these findings.

Response of the DMN showed a different pattern of activations from the one found for regions of the caregiving network. We included the DMN in our preregistered ROIs to assess the caregiving network in comparison with another well-characterized network known to be sensitive to self-related processing (*Andrews-Hanna et al., 2014*; *Northoff et al., 2006*; *Peer et al., 2015*) and to pinpoint the effects of social moments on this network. The DMN provides a useful comparison as it is thought to sustain the sense of self and is sensitive to self-other distinction across numerous types of stimuli (*Davey et al., 2016*; *Salomon et al., 2009*). As expected, the DMN showed a significant effect of Self vs. Other; however, contrary to previous studies (*Andrews-Hanna et al., 2014*; *Northoff et al., 2006*; *Salomon et al., 2014*), it showed higher BOLD responses to the Other condition. This is likely due to the mother's orientation towards external stimuli and possibly as a result of averaging the different regions of the DMN, which have different functional selectivity to self-related stimuli (*Araujo et al., 2015*; *Davey et al., 2016*; *Northoff et al., 2006*; *Salomon et al., 2014*). Still, while the caregiving network showed decreased activation under OT to social stimuli, no such effects emerged for the DMN. In addition, the DMN did not display consistent temporal patterns in the WSC. Additional analysis of the visual network, a task positive network that was activated by the visual stimulation, similarly showed no differential response for social stimuli under OT, highlighting

the specificity of our findings to the maternal caregiving network and to mother-infant social moments.

Limitations of the study include the relatively small number of participants and the relatively high attrition rate, which partly relate to the fact that mothers were imaged twice within a 2-week span during the postpartum and if mothers could not schedule the next meeting or one scan had technical problems the participant was excluded from the study. Similarly, inclusion of more controlled stimuli alongside the ecologically valid ones could have shed further light on the mother's caregiving network. Despite these limitations, our study introduces a novel ecologically-valid paradigm, examines pre-registered ROIs, and integrates multi-measure methodologies, including brain imaging, social behavioral coding, OT administration, and hormonal analysis to expand knowledge on the mother's neural response to social moments.

Characterization of the human mother's brain requires much further study. Whereas our study examined the neural responses of postpartum mothers to social and non-social caregiving experiences, future research may target other caregivers, such as fathers, grandparents, or childcare providers. Similarly, we imaged mothers who are raising infants in typical mother-father families and other family constellations, such as single or same-sex parents require further research.

Future work could rigorously test the cross-generational transmission of human sociality through longitudinal studies that examine linkage between a mother's neural response to social moments with her infant, versus non-social caregiving experiences, and her child's future social-emotional development, psychological well-being, or indices of brain maturation and neural response to social and affiliative cues. Another window into the cross generational transmission is by targeting high-risk conditions known to impact children's social competencies. Neural activations in the maternal brain to infant stimuli are attenuated in conditions such as poverty (*Kim et al., 2017*) or depression (*Kim et al., 2016*) and longitudinal studies show that disruptions in maternal synchronous caregiving predict later insensitivity of the child's brain to attachment cues (*Pratt et al., 2019*). However, a longitudinal study linking a mother's neural response in the postpartum with her child's later brain activations to social stimuli has not been conducted and such research may shed further light on how the mother's neural response to moments of social synchrony plays a particularly important role in tuning the child's brain to social cues. Finally, our findings can inform the construction of specific interventions that target maternal neural response to social interactions and aim to boost the salience and reward of the infant and the attachment relationship. Much further research is needed to fully understand the mechanisms that sustain consolidation of the caregiving network in health and pathology and describe how cultural practices, personal habits, and meaning systems shape the mother's neural response and are then transferred to the infant to cement the transmission of human cultural heritage.

## Materials and methods

### Participants

The initial sample included thirty-five postpartum mothers who were recruited through advertisements in various parenting online forums. Following recruitment, mothers underwent a brief phone screening for MRI scanning and postpartum depression using the Edinburgh Postnatal Depression Scale (*Cox et al., 1987*). Cutoff for joining the study was EPDS score of 8 and below (score above nine indicates minor depression). Next, mothers were invited to a psychiatric clinic to be tested by a psychiatrist prior to OT administration. During this visit, mothers were interviewed using the Structured Clinical Interview for the DSM-IV (SCID) to assess current and past psychiatric disorders. None of participants met criteria for a major or minor depressive episode during the perinatal period, 97% did not meet criteria for any diagnosable psychopathology, and 86% did not meet criteria for any diagnosable psychopathology disorder during their lifetime. All participants in the study were married, cohabitated with the infant's father, were of middle-or upper-class socioeconomic status, and completed at least some college.

Of the 35 participants, three did not complete a single scan (one due to medical problems and two due to claustrophobia). After examining the quality of the data six mothers were excluded due to excessive head movement artifacts (movements $\geq$ 3 mm). In additional three participants, we identified unexplained noise in the signal, found by contrasting the visual conditions vs rest. All nine

subjects were removed before analysis of the experimental effects. The final sample included twenty-three mothers (mean age = 28.8 years, SD = 4.7; EPDS mean score = 2.48, SD = 2.66) of 4–8 month-old infants (mean age = 5.78 months, SD = 1.25) who underwent scanning twice (46 scans). The study was approved by the Bar-Ilan University's IRB and by the Helsinki committee of the Sourasky medical center, Tel Aviv (Ethical approval no. 0161–14-TLV). All participants signed an informed consent. Subjects received a gift certificate of 700 NIS (~200 USD) for their participation in all four phases of the study (diagnosis, home visit, and two imaging sessions).

## Procedure

Following psychiatric diagnosis, the study included three sessions. In the first, families were visited at home, several episodes of mother–infant interactions were videotaped, and mothers completed self-report measures.

Several films were used as stimuli for the functional magnetic resonance imaging (fMRI) sessions. The videos depicted three typical situations distinguished by the amount of mother-infant social interaction and included: 1. *Unresponsive* condition (mother sitting next to the infant busy with her cellphone), 2. *Unavailable* condition (mother facing infant but not interacting socially), and 3. *Social* condition (mother engaging in a face-to-face peek-a-boo interaction). In all interactions mothers were instructed to sit next to their infants in the same distance and used standard toy and infant seat.

In the second and the third sessions, mothers participated in brain scanning at the Tel-Aviv Sourasky Medical Center. Mothers were instructed to avoid food intake and breastfeeding two hours before arrival. Before each scan mothers received 24 IU of oxytocin or placebo intranasally in a randomized, placebo-controlled, double-blind, two-period crossover design. During each session, salivary samples for oxytocin were collected at three time-points: immediately after consent and before OT or Placebo administration, following OT or Placebo administration and before participants were taken for the fMRI scan, and after the scan. While in the scanner, mothers were presented with vignettes of individually-tailored stimuli of own mother–infant interactions and with fixed control stimuli of unfamiliar mother and infant interactions. On average, 14 days elapsed between the two scans (SD = 11.67, mode = 7, median = 7), that were both scheduled for the morning hours (07:30-12:00). Study procedure is presented in *Figure 6*.

## Oxytocin administration and salivary oxytocin collection and measurement

Mothers were asked to self-administer 24 IU of either oxytocin (Syntocinon Nasalspray, Novartis, Basel, Switzerland; three puffs per nostril, each containing 4 IU) or placebo. The placebo was custom-designed by a commercial compounding pharmacy to match drug solution without the active ingredient. The same type of standard pump-actuated nasal spray was used for both treatments. Three saliva samples were collected by passive drooling into a tube prior to inhaling oxytocin or placebo (baseline); 10–15 min after administration (post administration); and at the end of fMRI session (recovery). All samples were kept chilled and stored at −20°C. The concentration of OT was determined by Cayman-OT ELISA kit (Cayman Chemicals, Ann Arbor, Michigan, USA). Consistent with prior research we used ELISA (enzyme-linked immunosorbent assay), a method commonly used for analyzing hormones in saliva (*Gordon et al., 2013*; *Rassovsky et al., 2019*). In order to prepare the sample for measurement, samples underwent the following: 1. Freeze-thaw three cycles: freeze at −80°C and thaw at 4°C to precipitate the mucus; 2. Centrifugations at 1500 g (4000 rpm) for 30 min; and 3. The supernatant was transferred into clean tube, and stored at −20°C until assayed. Concentration of OT in these samples was determined according to the manufacturer's kit instructions. The inter-assay coefficients of samples and controls were less than 18.7%, in the rage reported by the manufacture.

## MRI scans

### Data acquisition

Magnetic Resonance Imaging (MRI) data was collected using a 3T scanner (SIEMENS MAGNETOM Prisma syngo MR D13D, Erlangen, Germany) located at the Tel Aviv Sourasky Medical Center. Scanning was conducted with a 20-channel head coil for parallel imaging. Head motion was minimized by

padding the head with cushions, and participants were asked to lie still during the scan. High-resolution anatomical T1 images were acquired using magnetization prepared rapid gradient echo (MPRAGE) sequence: TR = 1860 ms, TE = 2.74 ms, FoV = 256 mm, Voxel size = 1×1 × 1 mm, flip angle = 8 degrees. Following, functional images were acquired using EPI gradient echo sequence. TR = 3000 ms, TE = 35 ms, 44 slices, slice thickness = 3 mm, FOV = 220 mm, Voxel size = 2.3×2.3 × 3 mm, flip angle = 90 degrees. In total 170 volumes were acquired over the course of the 'maternal condition' paradigm. Visual stimuli were displayed to subjects inside the scanner, using a projector (Epson PowerLite 74C, resolution = 1024 × 768), and were back-projected onto a screen mounted above subjects' heads, and seen by the subjects via an angled mirror. The stimuli were delivered using 'Presentation' software (http://www.neurobs.com).

### fMRI task
The three maternal conditions paradigm and fMRI sequence began about 50 min after intranasal Oxytocin/Placebo administration. During scanning, participants observed six naturalistic videos of 60 s each depicting themselves interacting with their babies ('self' condition) and similar videos of an unfamiliar standard mother interacting with her baby ('other' condition). Between videos a fixation of a black cross over a gray background was presented. Fixation duration was alternated between 15 and 18 s. The order of conditions was counterbalanced across subjects and scans. While in the scanner mothers were asked to watch the movies attentively. Video clips were played using VLC media-player (version 2.2 for windows, VideoLAN, France).

## fMRI analysis
### Data preprocessing
Data preprocessing and data analysis were conducted using BrainVoyager QX software package 20.6 (Brain Innovation, Maastricht, The Netherlands, RRID: SCR_013057) (Goebel et al., 2006). The first three functional volumes, before signal stabilization, were automatically discarded by the scanner to allow for T1 equilibrium. Preprocessing of functional scans included 3D motion correction, slice scan time correction, spatial smoothing by a full width at half maximum (FWHM) 6 mm Gaussian kernel, and temporal high-pass filtering. The functional images were then manually aligned and co-registered with 2D anatomical images and incorporated into the 3D datasets through trilinear interpolation. The complete dataset was normalized into MNI (Montreal Neurological Institute) space (Evans et al., 1994).

### Whole brain analysis
Multi-subject general linear models (GLM) were computed with random effects, with separate subject predictors, in which the various blocks (videos or fixation) were defined as predictors and convoluted with a standard hemodynamic response predictor. Following, a whole brain, three factors (Maternal Condition × Self-Other × PBO-OT) repeated measures ANOVA was performed. Whole brain maps were created and corrected for false discovery rate (FDR) of q < 0.050 (Benjamini et al., 1995). For visualization of results, the group contrasts were overlaid on a MNI transformed anatomical brain scan.

In order to examine the origin of the significant 'maternal condition' factor main effect, we computed group FDR corrected whole brain maps of the contrasts: social ≥ unavailable; social ≥ unresponsive; unavailable ≥ unresponsive. Effects in areas that were not included in our a priori Regions of Interest were reported for descriptive purposes only.

### Regions-of-interest preregistration and analysis
Region-of-interest (ROI) analysis was conducted on eight preregistered bilateral defined ROIs (https://osf.io/mszqj/?view_only=0daf10c02c984ead8929452edf44e550) including the amygdala, anterior cingulate (ACC), anterior insula, hippocampus/ parahippocampal gyrus, Temporal pole, VTA, NAcc (all together defined as the 'maternal caregiving network') and the DMN. ROI selection was a priori based on theory and literary meta-reviews (Abraham et al., 2016; Lindquist et al., 2016), and on pilot study of 4 subjects that completed similar paradigm and were not included in the current study. ROIs were defined functionally and anatomically, verified and validated by human brain database platforms: Talairach Daemon (Lancaster et al., 2000) and Neurosynth

(*Yarkoni et al., 2011*), registered at the Open Science Framework prior to data analysis (*OSF, 2020*) and transformed into MNI space (*Appendix 1—figure 6*.). In addition, visual network was defined based on the Glasser atlas and analyzed as 'task positive' control to the maternal caregiving network (see Appendix 1 for details).

Beta weights were extracted from ROIs and analyzed with a $7 \times 3 \times 2 \times 2$ (*Maternal caregiving network ROIs $\times$ Maternal Condition $\times$ Self-Other $\times$ PBO-OT*) repeated measures ANOVA using JASP (Version 0.9 for windows, JASP Team, 2018, RRID: SCR_015823). Thus, allowing to investigate main effects of oxytocin administration, and stimulus type and their interactions. In order to further examine the origin of main effects and interactions, simple effect analyses, Scheffé and Bonferroni post hoc tests were conducted. Null effects were analyzed using Bayesian methods (JASP) using default prior (*Keysers et al., 2020*). $BF_{incl}$ is calculated for repeated measures ANOVA's using the Baws factor approach across all matched models.

### Within subject correlation (WSC)

In order to test the consistency of temporal patterns during different conditions, between oxytocin and placebo, we calculated a within-subject correlation (WSC) for each subject in each condition in each ROI. The WSC is the Pearson's correlation for the BOLD time course of each condition (e.g. 'Self-Social') in a specific ROI, between the OT and PBO runs. Thus for each participant the correlation indicated how similar was the dynamics of the response in a specific ROI while watching an identical movie clip under the OT or the PBO conditions. To test differences between the temporal activation consistencies of the preregistered ROIs in the three conditions, a $7 \times 3 \times 2$ (maternal caregiving *ROI $\times$ Maternal Condition $\times$ Self-other*) repeated measures ANOVA and post hoc tests were conducted. The same analysis was performed for the DMN. Next, we conducted Bayesian methods for the analyses of null effects.

### Behavioral coding

Mirco-level synchrony of the three conditions. To verify that the Social condition indeed was characterized by greater synchrony all video vignettes mothers observed in the scanner were micro-coded by trained coders on a computerized system (Mangold- Interact, Arnstorf, Germany, RRID: SCR_019254) in 3 s frames. Consistent with much prior research in our lab (*Feldman and Eidelman, 2007*; *Feldman and Eidelman, 2003*), four non-verbal categories of infant behavior were coded and each category included a set of mutually exclusive codes (an 'uncodable' code was added to each category): Affect (excitement, positive, neutral, medium-fussing, negative, relief after pressure), Gaze (joint attention, to mother's face, to object or body part, scanning, gaze aversion), Vocalization (no vocalization, positive, neutral, regulatory, negative), and Movement (no movement, hand waving, leg kicking). Mother behavior was coded for Affect and Gaze. Synchrony was defined, consistent with our prior research, by conditional probabilities (infant in state A given mother in state A), indicating episodes when mother and infant were both in social gaze and shared positive affect (*Feldman and Eidelman, 2007*; *Granat et al., 2017*). In addition, mother-infant synchrony during a free interaction in the home environment was coded using the Coding Interactive Behavior (CIB) Manual (*Feldman, 1998*), a global rating system (see Appendix 1).

### Statistical analysis

For statistical analysis JASP (Version 0.9 for windows, JASP Team, 2018, RRID: SCR_015823), SPSS (SPSS statistics Version 25.0, IBM Corp. Armonk, NY) and R software (Version 3.5.3, R Core Team, 2017, Vienna, Austria, RRID: SCR_019096) were used.

## Acknowledgements

The study was supported by the Simms/Mann Foundation. The authors wish to thank Professor Talma Hendler for her valuable contribution.

## Additional information

### Funding

| Funder | Author |
|---|---|
| Simmons Family Foundation | Ruth Feldman |

The funders had no role in study design, data collection and interpretation, or the decision to submit the work for publication.

### Author contributions

Ortal Shimon-Raz, Conceptualization, Resources, Data curation, Formal analysis, Supervision, Funding acquisition, Visualization, Methodology, Writing - original draft, Project administration, Writing - review and editing; Roy Salomon, Conceptualization, Data curation, Software, Formal analysis, Supervision, Visualization, Methodology, Writing - original draft, Project administration, Writing - review and editing; Miki Bloch, Conceptualization, Resources, Software, Supervision, Methodology, Writing - original draft, Writing - review and editing, psychiatric evaluation of subjects; Gabi Aisenberg Romano, Conceptualization, Resources, Software, Visualization, Methodology, Writing - review and editing, psychiatric evaluation of subjects; Yaara Yeshurun, Software, Visualization, Methodology, Writing - review and editing; Adi Ulmer Yaniv, Orna Zagoory-Sharon, Software, Formal analysis, Visualization, Methodology, Writing - review and editing; Ruth Feldman, Conceptualization, Resources, Software, Formal analysis, Supervision, Funding acquisition, Visualization, Methodology, Writing - original draft, Writing - review and editing

### Author ORCIDs

Ortal Shimon-Raz https://orcid.org/0000-0002-6464-8370
Roy Salomon https://orcid.org/0000-0002-6688-617X
Adi Ulmer Yaniv https://orcid.org/0000-0001-5865-8305
Ruth Feldman https://orcid.org/0000-0001-5048-1381

### Ethics

Human subjects: The study was approved by the Bar-Ilan University's IRB and by the Helsinki committee of the Sourasky medical center, Tel Aviv (Ethical approval no. 0161-14-TLV). All participants signed an informed consent.

### Decision letter and Author response

Decision letter https://doi.org/10.7554/eLife.59436.sa1
Author response https://doi.org/10.7554/eLife.59436.sa2

## Additional files

### Supplementary files

• Transparent reporting form

### Data availability

Raw, subject by subject, anonymized brain data (fMRI); group level data (e.g. unthresholded group maps on MNI template) and raw subject by subject data from the ROI analysis (csv and JASP files) are freely available. These files are uploaded to our OSF account (https://osf.io/mszqj/?view_only=0daf10c02c984ead8929452edf44e550) to allow full transparency of the data.

The following dataset was generated:

| Author(s) | Year | Dataset title | Dataset URL | Database and Identifier |
|---|---|---|---|---|
| Shimon-Raz O, | 2020 | Mother Brain is Wired for Social | https://osf.io/mszqj/? | Open Science |

| Salomon R, Bloch M, Romano GA, Yeshurun Y, Ulmer-Yaniv A, Zagoory-Sharon O, Feldman R | Moments | view_only=0daf10c02-c984ea-d8929452edf44e550 | Framework, mszqj |

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

## Appendix 1

### Results

ROI analysis in the maternal caregiving network
Self-other main effect

Differential responsivity to *Self-Other* conditions was found across the maternal caregiving network. Greater activation in response to the Self compared to Other-stimuli was found in the insula (Mean$_{self}$ = 0.43, SD$_{self}$ = 0.14; Mean$_{other}$ = −0.09, SD$_{other}$ = 0.18) and the ACC (Mean$_{self}$ = −0.08, SD$_{self}$ = 0.19; Mean$_{other}$ = −0.22, SD$_{other}$ = 0.22). In contrast the amygdala (Mean$_{self}$ = 0.09, SD$_{self}$ = 0.19; Mean$_{other}$ = 0.23, SD$_{other}$ = 0.20), TP (Mean$_{self}$ = 0.12, SD$_{self}$ = 0.23; Mean$_{other}$ = 0.22, SD$_{other}$ = 0.14), parahippocampal gyrus (Mean$_{self}$ = −0.03, SD$_{self}$ = 0.19; Mean$_{other}$ = 0.06, SD$_{other}$ = 0.13) the NAcc (Mean$_{self}$ = −0.19, SD$_{self}$ = 0.25; Mean$_{other}$ = −0.01, SD$_{other}$ = 0.22) were more activated during the Unfamiliar-Other compared to the Self- stimuli (*Appendix 1—figure 1*, *Appendix 1—table 1*).

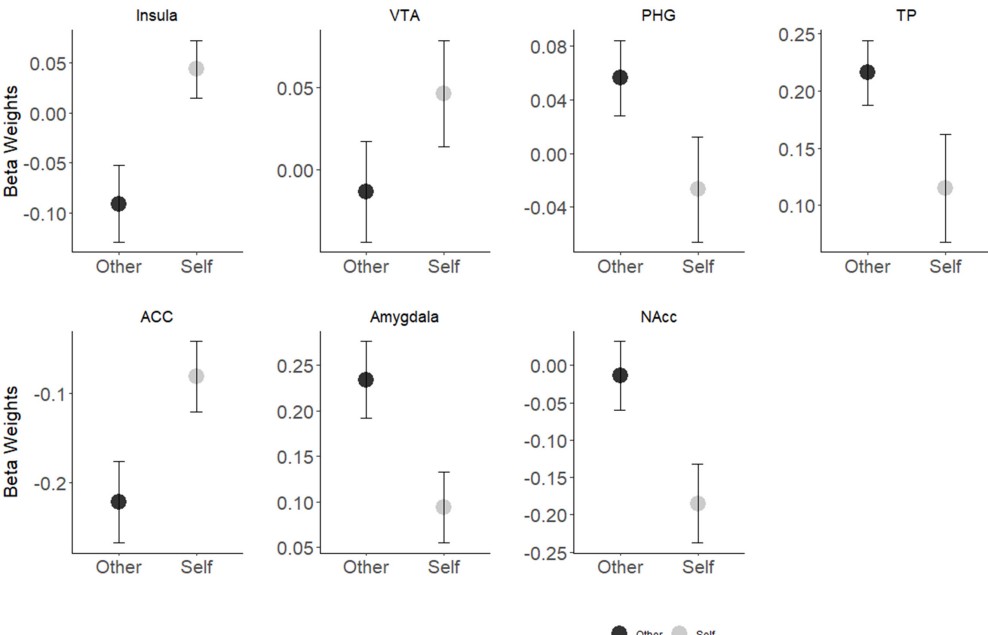

**Appendix 1—figure 1.** *Self-Other* main effect in seven preregistered ROIs. The insula and ACC showed greater activation in response to *self-* stimuli while greater activation in response to *other-* stimuli was found in the amygdala, TP, parahippocampal gyrus and in the NAcc. Differences in the VTA were not significant. Bars depict Standard error of the mean; NAcc, Nucleus accumbens; PHG, parahippocampal gyrus; VTA, ventral tegmental area.

**Appendix 1—table 1.** Results of significant Self-Other × ROI interaction effect within a repeated measures ANOVA (ROI × Maternal Condition × Self-Other × PBO- OT) separately for each of the preregistered ROIs.

post hoc analysis of significant interaction done separately for each of the preregistered ROIs. All results are Greenhouse-Geisser corrected. OT, oxytocin; PBO, placebo; ACC, anterior cingulate cortex; NAcc, nucleus accumbans; PHG, parahippocampal gyrus; TP, temporal pole; VTA, ventral tegmental area. *, p<.05; ψ, Bonferroni correction for multiple comparisons.

| | Maternal Caregiving Network | | | | | | |
|---|---|---|---|---|---|---|---|
| | Insula | ACC | TP | Amygdala | VTA | PHG | NAcc |
| *Self-Other main effect, df (1,22)* | | | | | | | |
| F score | 22.97 | 13.78 | 4.47 | 6.81 | 2.91 | 6.32 | 11.27 |

*Continued on next page*

*Appendix 1—table 1 continued*

| | **Maternal Caregiving Network** | | | | | | |
|---|---|---|---|---|---|---|---|
| | **Insula** | **ACC** | **TP** | **Amygdala** | **VTA** | **PHG** | **NAcc** |
| P | <0.001$^\Psi$ | 0.001$^\Psi$ | 0.046* | 0.016* | 0.102 | 0.020* | 0.003$^\Psi$ |
| Eta$^2$ | 0.51 | 0.39 | 0.17 | 0.24 | 0.12 | 0.22 | 0.34 |

## ROI analysis in the DMN
### Self-Other main effect

As expected, the DMN showed sensitivity to Self- Other distinctions, however it showed stronger BOLD activations to the Other compared to Self- stimuli (Mean$_{self}$ = 0.02, SD$_{self}$ = 0.17; Mean$_{other}$ = 0.08, SD$_{other}$ = 0.18) (*Appendix 1—figure 2*, *Appendix 1—table 2*) however Bayesian analysis did not support this effect (BF$_{incl}$ = 0.616).

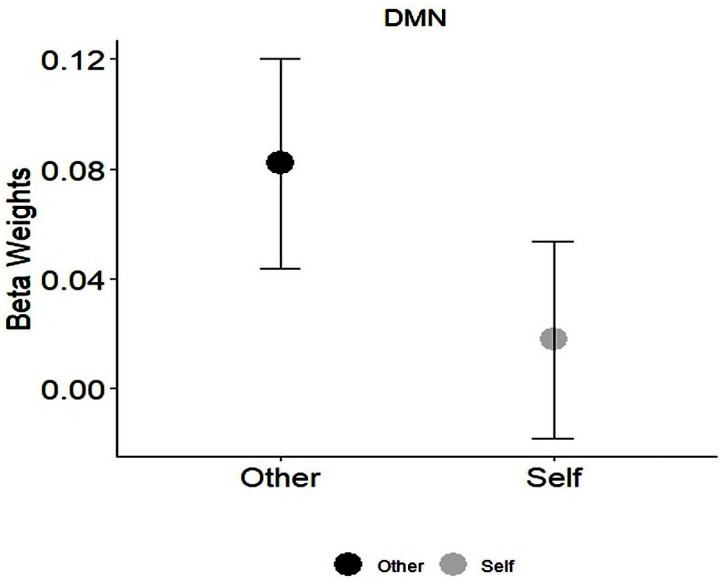

**Appendix 1—figure 2.** *Self-Other* main effect in the DMN.

**Appendix 1—table 2.** Results of Self-Other and Maternal Condition main effects within a repeated measures ANOVA (Maternal Condition × Self-Other × PBO- OT) in the DMN.
All results are Greenhouse-Geisser corrected. DMN, default mode network. *, p<.05.

| **DMN** | |
|---|---|
| *Self-Other main effect, df (1,22)* | |
| F score | 4.93 |
| P | 0.037* |
| Eta$^2$ | 0.18 |
| *Maternal Condition main effect, df (2,44)* | |
| F score | 1.18 |
| P | 0.32 |
| Eta$^2$ | 0.05 |

## Condition × PBO-OT interaction

In the DMN, no significant effect of *Condition × PBO-OT* interaction was found [F (1.93, 42.56) =2.74, p=0.077] (*Appendix 1—figure 3*). Bayesian analysis indicated moderate evidence for the absence of such significant interaction effect (BF$_{incl}$ = 0.809).

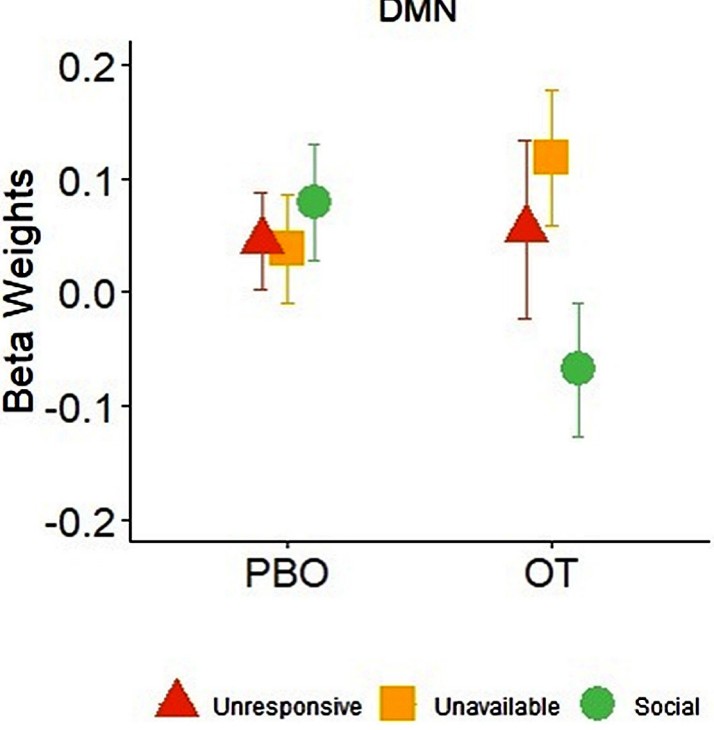

**Appendix 1—figure 3.** Interaction effects of Condition × PBO-OT in the DMN. Bars depict Standard error of the mean. PBO, placebo; OT, Oxytocin; DMN, Default mode network.

## Visual network

In order to examine whether our results originated from a systemic response of the whole brain, we further conducted ROI analysis to a task negative visual network as appears in Glasser atlas (Glasser areas 1,5,6,8,9,25,26) and includes BA 17,18,19. three factors repeated measures ANOVA *Maternal Condition × Self-Other × PBO- OT* was performed. No significant effects were found in the network (*Appendix 1—tables 3*, *4*.) nor in each of the areas separately.

**Appendix 1—table 3.** Results of 3 factors repeated measures ANOVA (Maternal Condition × Self-Other × PBO- OT) in the visual system.
All results are Greenhouse-Geisser corrected. OT, oxytocin; PBO, placebo.

|  | df | F score | P | Eta² |
|---|---|---|---|---|
| *PBO-OT* main effect | 1,22 | 0.003 | 0.956 | 0.000 |
| *Maternal Condition* main effect | 1.849, 40.672 | 1.037 | 0.359 | 0.045 |
| *Self-Other* main effect | 1,22 | 2.153 | 0.156 | 0.089 |
| *PBO-OT × Maternal Condition* interaction | 1.923, 42.312 | 2.251 | 0.120 | 0.093 |
| *PBO-OT × Self-Other* interaction | 1,22 | 0.09 | 0.926 | 0.000 |
| *Maternal Condition × Self-Other* interaction | 1.759, 38.698 | 1.610 | 0.215 | 0.068 |
| *PBO-OT × Maternal Condition × Self-Other* interaction | 1.647,36.226 | 0.362 | 0.658 | 0.016 |

**Appendix 1—table 4.** Results of analysis of effects within a 3 factors Bayesian repeated measures ANOVA (Maternal Condition× Self-Other× PBO-OT) in the visual system.
BFincl is calculated using the Baws factor approach across all matched models.

| Effects | P (incl) | P (incl, data) | BF$_{incl}$ |
|---|---|---|---|
| *PBO-OT* | 0.263 | 0.115 | 0.132 |
| *Maternal Condition* | 0.263 | 0.212 | 0.077 |
| *Self-Other* | 0.263 | 0.071 | 0.271 |
| *PBO-OT× Maternal Condition* | 0.263 | 0.003 | 0.295 |
| *PBO-OT × Self-Other* | 0.263 | 0.005 | 0.177 |
| *Maternal Condition × Self-Other* | 0.263 | 0.002 | 0.154 |
| *PBO-OT× Maternal Condition × Self-Other* | 0.053 | 2.021e −6 | 0.128 |

## WSC- DMN

In the DMN, no significant effect of maternal condition [F (1.97, 43.42)=0.87, p=0.42], Self-Other [F (1, 22)=0.06, p=0.81] or interaction between them [F (1.89, 41.66)=0.34, p=0.70] was found (*Appendix 1—figure 4*). Bayesian two-way repeated measures ANOVA (*Condition × Self-Other*) showed strong evidence for the absence of significant difference in WSC between the maternal conditions in the DMN (BF = 0.097).

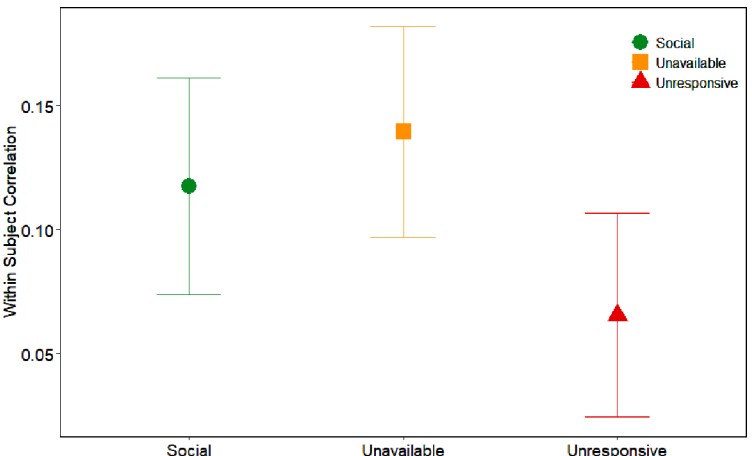

**Appendix 1—figure 4.** DMN results of WSC comparison between conditions across the maternal brain. Contrary to the Maternal brain network, no significant effect was found in the DMN (BF = 0.097). Bars depict Standard error of the mean. DMN, Default mode network.

## Brain-Behavior coupling

To test out brain–behavior coupling in the Social condition, we computed Pearson's correlations between ROIs activation to the *Self-Social* condition under placebo (representing the mother's brain response under natural circumstances), and the global coding of 'mother-infant synchrony' measured during a free interaction in the home environment.

Mother-infant synchrony showed significant positive correlations with activity in the bilateral insula, ACC and VTA, under PBO indicating that more synchronous mothers exhibited greater activation in these areas to videos depicting themselves interacting with their infants in a social peek-a-boo game (*Appendix 1—figure 5*). Mother-infant synchrony was also positively correlated with activity in the VTA to the *Other-Social* condition under PBO ($r_p$ = 0.449, p=0.03). Mother-infant

synchrony was unrelated to neural activity during the 'unavailable' or 'unresponsive' conditions, highlighting the social context as the only one yielding brain-behavior coupling. It is important to note, however, that these correlations should be treated with caution as they are based on a relatively low number of participants.

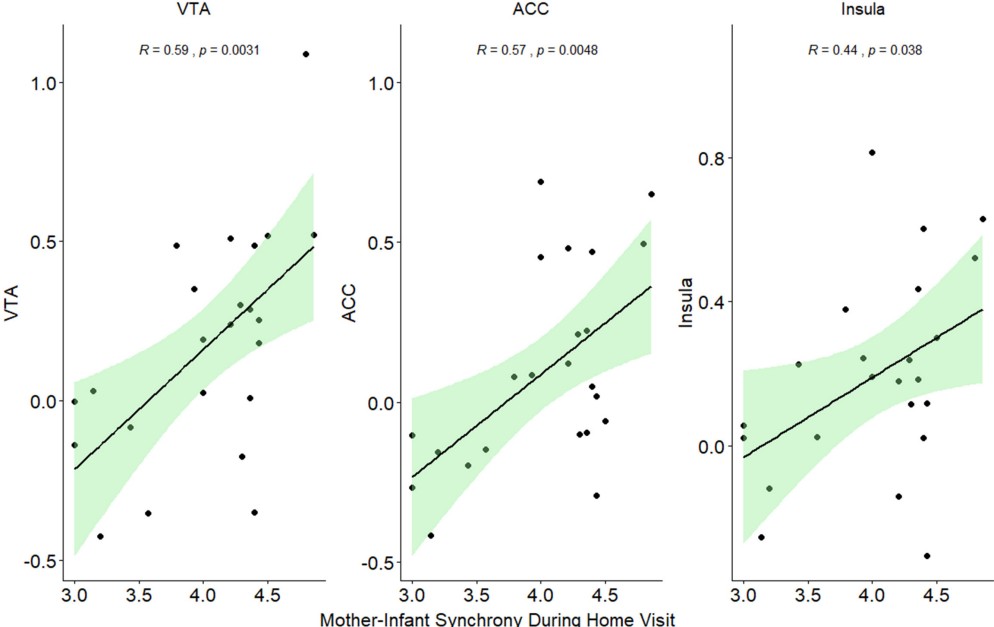

**Appendix 1—figure 5.** Regression lines of significant correlations between mother-infant synchrony during a free-play interaction at home visit and activation in the VTA, insula and ACC under 'Social' maternal condition and placebo. VTA, ventral tegmental area; ACC, anterior cingulate.

Bayesian Pearson's correlations indicated strong evidence for the correlation of mother-infant synchrony with activation to *Self - social* in the ACC (BF = 10.965) and in the VTA (BF = 15.783). No such evidence was found for correlations between synchrony and Insula activation to self-positive (BF = 1.980), and VTA to other positive (BF = 2.277).

In addition, infant social engagement, coded globally from the mother-infant interaction in the home ecology and assesses the degree of positivity and initiation the infant display during play showed a positive correlation with TP and VTA response to *Self* and *Other* social stimuli under placebo. However, Bayesian analyses demonstrated weak to moderate evidence for these correlations (*Appendix 1—table 5*).

**Appendix 1—table 5.** Significant Pearson's correlation and Bayesian results between infants' engagement and brain activation to social condition.

|  | Pearson's r | P | BF |
|---|---|---|---|
| TP Self Positive PBO | 0.521 | 0.011 | 5.525 |
| TP Other Positive PBO | 0.472 | 0.023 | 2.957 |
| VTA Self Positive PBO | 0.438 | 0.037 | 2.027 |
| VTA Other Positive PBO | 0.426 | 0.043 | 1.792 |

To the potential associations between maternal brain activation and infant temperament, two behavioral parameters of infants' toy exploration and affect were correlated with maternal brain response. As seen in the attached table, we found no significant correlations (*Appendix 1—table 6*).

**Appendix 1—table 6.** Pearson's correlations of infants affect and toy exploration with maternal brain response to self and other social condition under placebo.

| ROI | Infants affect (emotionality) | | | | Toy exploration (attention regulation) | | | |
|---|---|---|---|---|---|---|---|---|
| | Pearson's r | p | Upper 95% CI | Lower 95% CI | Pearson's r | p | Upper 95% CI | Lower 95% CI |
| ACC *Self* | 0.206 | 0.346 | 0.57 | −0.225 | −0.354 | 0.098 | 0.069 | −0.668 |
| ACC *Other* | −0.025 | 0.908 | 0.391 | −0.433 | −0.236 | 0.278 | 0.195 | −0.591 |
| Amygdala *Self* | −0.023 | 0.916 | 0.393 | −0.431 | −0.358 | 0.093 | 0.064 | −0.671 |
| Amygdala *Other* | 0.040 | 0.858 | 0.445 | −0.379 | −0.516 | 0.012 | −0.132 | −0.765 |
| Insula *Self* | −0.005 | 0.981 | 0.408 | −0.417 | −0.182 | 0.405 | 0.248 | −0.553 |
| Insula *Other* | 0.006 | 0.979 | 0.417 | −0.407 | −0.322 | 0.135 | 0.104 | −0.648 |
| NAcc *Self* | 0.005 | 0.983 | 0.416 | −0.408 | −0.154 | 0.483 | 0.276 | −0.532 |
| NAcc *Other* | −0.053 | 0.810 | 0.367 | −0.455 | −0.078 | 0.725 | 0.346 | −0.475 |
| PHG *Self* | 0.043 | 0.847 | 0.447 | −0.376 | 0.114 | 0.605 | 0.503 | 0.313 |
| PHG *Other* | −0.077 | 0.727 | 0.346 | −0.474 | −0.066 | 0.765 | 0.356 | −0.465 |
| TP *Self* | 0.049 | 0.823 | 0.452 | −0.37 | −0.063 | 0.776 | 0.359 | −0.463 |
| TP *Other* | 0.049 | 0.823 | 0.452 | −0.37 | −0.313 | 0.145 | 0.113 | −0.643 |
| VTA *Self* | 0.020 | 0.926 | 0.429 | −0.395 | −0.200 | 0.360 | 0.231 | −0.566 |
| VTA *Other* | −0.013 | 0.953 | 0.401 | −0.423 | −0.411 | 0.051 | 0.001 | −0.704 |

## Methods

### Assessing mother-infant synchrony in the natural ecology

Mother-infant synchrony in the home environment was coded using the Coding Interactive Behavior (CIB) Manual (*Feldman, 1998*). The CIB is a global rating system for adult– child interactions that includes 42 scales that aggregate into theoretically meaningful constructs. The CIB is well-validated with good psychometric properties and has been extensively used across the world in research on health and high-risk population (*Feldman, 2012*).

### Mother-infant synchrony construct

consistent with prior research, the synchrony construct of the CIB includes the codes of dyadic reciprocity, mutual adaptation, and fluency, and these codes were averaged into a 'mother-infant synchrony' construct. Coding was conducted by a trained coder blind to any other information and inter-rater reliability averaged 95% (k = 0.87).

### Infant social engagement

we used the CIB coding scheme for assessing child social behavior (social engagement) during the naturalistic mother-child interaction. The child engagement construct is the average of the following scales: infant social initiation, infant alert, infant positive affect and infant vocalization during the interaction. Cronbach's alpha coefficient was calculated for the four subscale items indicating good internal consistency reliability (k = 0.82).

### Infant temperamental dispositions

While we regrettably do not have a direct measure of infant temperament (typically a self-report by the mother), to the potential associations between maternal brain activation and infant temperament, we analyzed the 'infant alone' condition, a two-minute video vignette in which the infants

played alone with age-appropriate standard toys. Such setting is often used in the developmental literature to measure temperament (*Goldsmith and Rothbart, 1996*) and to assess infant persistent attention, sustained exploration, and affect, which are marker of emotionality and regulation, the core features of infant temperament (*Rothbart, 1981*). We micro-coded on a second-by-second level two behavioral parameters of toy exploration and affect, each on a scale of 1 (low) to 5 (high): sustained attention/persistent exploration (index of regulation) and affect (index of emotionality). The *attention/exploration scale (attention regulation)* was coded as follows: 1 = no interest in the toy; 2 = holding toy + no gaze; 3 = mouthing; 4 = manipulation of toy + short gaze; 5 = sustained exploration with focused attention. The *affect scale (emotionality)* was coded as follows: 1 = very negative – crying; 2 = negative – fussing; 3 = neutral; 4 = positive; 5 = very positive. Average score of each parameter was calculated for each infant.

### Regions-of-interest preregistration

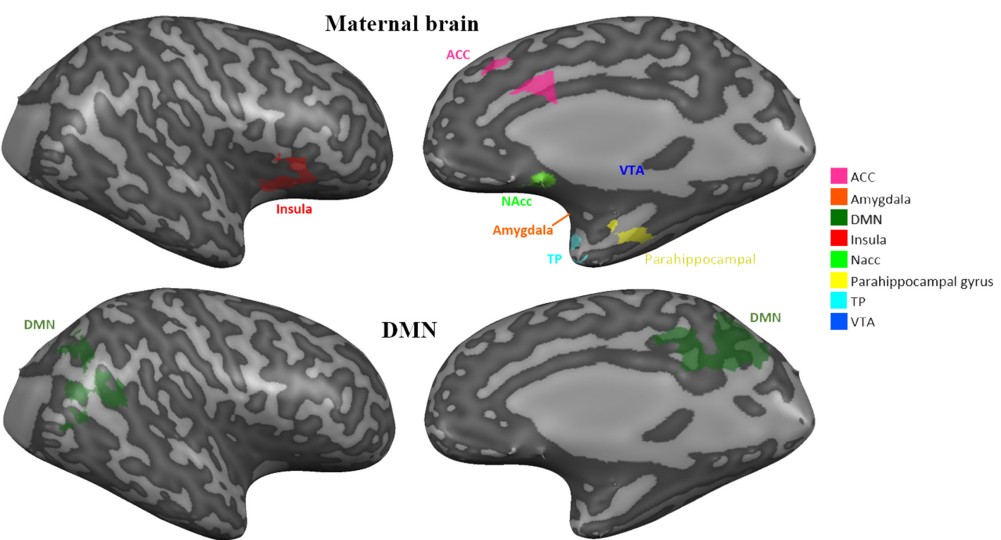

**Appendix 1—figure 6.** The 'Maternal caregiving network' and the DMN. eight preregistered ROI's laid on right hemisphere. ACC, Anterior cingulate; DMN, Default mode network; NAcc, Nucleus accumbens; TP, Temporal pole; VTA, ventral tegmental area.

