## [Decision Letter]

**Acceptance summary:**

The authors use fMRI to assess maternal brain activity in response to watching videos of mothers interacting with their infants compared to watching videos of other mothers/infants following intranasal oxytocin administration or a placebo. The videos were recorded in the mothers' own homes and involved three distinct kinds of interactions: social, unavailable, and unresponsive.

**Decision letter after peer review:**

Thank you for submitting your article "Mother Brain is Wired for Social Moments" for consideration by *eLife*. Your article has been reviewed by 2 peer reviewers, and the evaluation has been overseen by a Dr. Shackman as the Reviewing Editor and Dr. Büchel as the Senior Editor.

The Reviewing Editor has drafted this decision to help you prepare a revised submission.

Summary:

The authors use fMRI to assess maternal brain activity in response to watching videos of mothers interacting with their infants compared to watching videos of other mothers/infants following intranasal oxytocin administration or a placebo. The videos were recorded in the mothers' own homes and involved three distinct kinds of interactions: social, unavailable, and unresponsive.

The Reviewers and I saw several strengths to the report:

• This is an interesting manuscript examining an interesting topic. I read it with great interest.

• All in all, the findings are sound.

• The fact that this study was pre-registered is a major strength of this manuscript.

• The manuscript is well written and the study is novel and well-designed

Essential revisions:

• Conceptual Clarity. Both Reviewers noted that the concept of 'temporal engrams ' is unclear, as is its link to interoception and the brain

• Conceptual Precision. With all due respect to the authors, I find the terms "maternal brain," "social brain," etc. problematic. First, there is only one brain-the notion of "different brains" invokes the outdated triune brain concept. Second, talking about "the maternal brain," "social brain," and what have you is problematic because it implies a mutually exclusive set of brain regions. Yet the regions outlined herein are all equally likely to be involved in maternal behavior, social behavior, emotion, value-based decision-making etc. Finally, this terminology causes a form of essentialism that is problematic for scientists and lay people alike. Given that a paper like this will no doubt get some attention from the press, it seems all the more problematic to refer to the neural processes that are potentiated by OT following birth and that respond to an infants' cues as "the maternal brain." If you have postpartum depression is there something wrong with your maternal brain? If you don't give birth to your child or rear them from infancy, is your maternal brain deficient?

• Conceptual Framing: Model vs. Data. The biggest issue is the mismatch between the way in which the study is conceptually framed/interpreted and the actual data and results. The study is framed around the impacts of mother-infant synchrony effects on the infant, and the long-term consequences of those effects on the infant, whereas the data actually measured are impacts of mother-infant synchrony on the mother's own brain. If the authors prefer to keep the framing of the paper as it currently stands, then I think data that more directly assesses parenting behaviors of the parents, or temperament of their children, is needed. Do the authors have any additional data about the actual parenting behaviors of the parents, or temperament of their children, which might relate to the maternal brain activity?

• Condition Confounds. The greater reliability in brain activity between sessions for social v. non-social stimuli seems like an important confound. Can the authors be sure that greater reliability in activity between sessions for social behavior is not driven by greater similarity in stimulus-driven properties (e.g., peek-a-boo in both social conditions but heterogeneous behavior in control conditions)?

• Attrition. 12/35 (~34%) is a rather high attrition rate and raises important concerns about experimenter degrees of freedom and selective attrition. This needs to be carefully addressed and acknowledged as a limitation.

• Interpretation. Both Reviewers commented on this…

– The authors don't sufficiently deal with the fact that OT administration specifically *reduces* activity within a network that 'codes for' maternal behavior during social behavior. They argue that this indicates that the mother is in an anxiolytic state, but this seems a bit speculative-wouldn't she be in an anxiolytic state regardless of the stimuli being observed? It is true that OT is thought to have an "anxiolytic" effect in rat models, insofar as it diminishes neophobia of pups amongst nulliparous females. When it does so in rats, it changes the functional connectivity within the maternal network in terms of which brain regions are involved. Yet this does not necessarily describe why human mothers would have reduced activity within the general network supporting maternal behavior under OT. Clearly there is some effect of OT, but the interpretation of what a reduced BOLD signal means is far from clear. My personal take is that we still know too little about what the BOLD signal actually indexes to make too many inferences about what an overall reduction of BOLD within a network under pharmacological manipulation actually means, psychologically.

– The rationale and interpretation of the within-subject correlation (WSC) is unclear. If oxytocin is expected to upregulate neural responses within the maternal care network, in response to social stimuli, then wouldn't one expect there to be a lower WSC across the oxytocin and placebo conditions for the social condition? My apologies if I am missing something obvious here; if so, please provide a more explicit explanation within the main text.

[Editors' note: further revisions were suggested prior to acceptance, as described below.]

Thank you for resubmitting your article "Mother Brain is Wired for Social Moments" for consideration by *eLife*. Your revised article has been reviewed by 2 peer reviewers, and the evaluation has been overseen by Ds. Shackman as the Reviewing Editor and Dr. Büchel as the Senior Editor.

The Reviewing Editor has drafted this to help you prepare a revised submission.

Summary:

The authors use fMRI to assess maternal brain activity in response to watching videos of mothers interacting with their infants compared to watching videos of other mothers/infants following intranasal oxytocin administration or a placebo. The videos were recorded in the mothers' own homes and involved three distinct kinds of interactions: social, unavailable, and unresponsive.

The Reviewers and I saw several strengths to the revised report:

• This is an interesting manuscript examining an interesting topic. I read it with great interest.

• All in all, the findings are sound.

• The fact that this study was pre-registered is a major strength of this manuscript.

• The manuscript is well written and the study is novel and well-designed

Essential revisions:

Both Reviewers remained concerned about the conceptual framing and underscored the need to temper the claims to better align with the approach and results (i.e. avoid "overselling")…

• Remove the temporal engram concept from the manuscript entirely.

– The only thing I still have concerns about is the temporal engram argument. It is not clear to me that the authors have demonstrated the existence of a "temporal engram" for the infant – at least in their definition – using their WSC analyses.

– As I understand their logic, the point of a temporal engram is that it ""engrave(s)" a temporal representation of the infant in the maternal caregiving network and hypothesized that synchrony and its dyad-specific rhythms may build and amplify temporal patterns in the mother's brain."

– However, the authors' findings do not support this argument – they show WSC for the social condition but no differentiation by self v. other, meaning that the network thought to be involved in representing social stimuli shows more reliable activity to social stimuli than non-social stimuli across instances, which is not in and of itself surprising.

– Likewise, they do not show specific activity for the "partners presence" (i.e., the infant).

– I am also still not convinced that the greater reliability is not a confound of stimulus properties. I understand that they saw the same stimulus at time 1 and time 2, but this does not solve the problem that there could be more randomness in neural responses to videos of unresponsive mothers because these videos are more boring, less salient, and generally entrain neural processes to a lesser degree.

– I'm certainly open to the interpretation that greater WSC to social v non-social stimuli is meaningful in some important way – but I think the authors are overselling it by trying to infer that this is a 'temporal engram.' My recommendation would be that they remove the temporal engram concept from the manuscript entirely.

• I think the framing of the paper needs to be altered so that the results are not misinterpreted; this would include changes in the abstract, introduction and Discussion sections to be more explicit the study's findings and being careful to not go beyond the data.

– There appears to be some modest support for the larger framework of the paper, with the mother's brain responses within particular nodes of the human caregiver network exhibiting modest relations with infant social engagement. Similarly, the mother's brain responses within particular nodes also exhibited some relations to mother-infant synchrony.

– Yet the more crucial analyses were not significant: no brain regions within the human caregiving network exhibited activity that was predictive of infant temperament, and moreover, mother-infant synchrony was not related to the infant's temperament.

[Editors' note: further revisions were suggested prior to acceptance, as described below.]

Thank you for submitting your article "Mother Brain is Wired for Social Moments" for consideration by *eLife*. Your article has been reviewed by 1 peer reviewers, and the evaluation has been overseen by Dr. Shackman as the Reviewing Editor and Dr. Büchel as the Senior Editor.

I'm very pleased to accept your report for publication pending receipt of a revision that addresses 3 key modifications.

Summary:

The authors use fMRI to assess maternal brain activity in response to watching videos of mothers interacting with their infants compared to watching videos of other mothers/infants following intranasal oxytocin administration or a placebo. The videos were recorded in the mothers' own homes and involved three distinct kinds of interactions: social, unavailable, and unresponsive.

The Reviewers and I saw several strengths to the revised report:

• This is an interesting manuscript examining an interesting topic. I read it with great interest.

• All in all, the findings are sound.

• The fact that this study was pre-registered is a major strength of this manuscript.

• The manuscript is well written and the study is novel and well-designed

• The data are interesting and certainly novel. The paradigm itself is impressive and I don't know of much other work that has tested these questions by combining scanning, oxytocin (OT) administration and naturalistic behavior.

Essential revisions:

Both Reviewers remain moderately concerned about the conceptual framing. They re-emphasized the need to temper the claims to better align with the approach and results (i.e. avoid "overselling") and to sketch out the most important challenges for the future

• A reviewer notes, My primary concern with this article remains that the authors appear to go beyond the data in the overall framing of the paper, specifically, their claims about "cross-generational transmission". In particular, the last line of their most recent abstract reads: "Findings describe how mother's brain compiles and amplifies these precious social moments to generate dyad-specific patterns that initiate the cross-generational transmission of human sociality." Additionally, the last two sentences of the first paragraph of the Discussion section: "Across mammalian species, the cross-generation transmission of sociality is initiated by the consolidation of the maternal neural network from which, through caregiving behavior, a similar network is sculpted in the infant's brain (Feldman, 2020; Numan, 2020). Our study uniquely tests the response of the human caregiving network in the maternal brain to these social moments, versus other moments of non-social mother-10infant presence, to shed further light on the cross-generation transmission of human sociality". I emphasized this issue in the most recent round of review. However, the authors unfortunately seem to have missed the central point of my comments.

• Based on a discussion with the Reviewers, we collectively recommend the following changes to the manuscript:

1. Eliminate the phrase "cross-generational transmission" (and similar) from the Abstract

2. Modify the Discussion, revising the sentence in question to read something like: "Our study uniquely tests the response of the human caregiving network in the maternal brain to these social moments, versus other moments of non-social mother-infant presence, to shed further light on how human mother brains may change following birth and vary with the caregiving experience with one's infant."

3. Provide a brief, but explicit discussion of ways in which future work could rigorously address "cross-generation transmission".

---

## [Author Response]

Essential revisions:• Conceptual Clarity. Both Reviewers noted that the concept of 'temporal engrams ' is unclear, as is its link to interoception and the brain.

We modified the introduction and provided a fuller explanation on the construct of 'temporal engrams'.

On page 7 we added:

"Finally, we explored the existence of "*temporal engrams*" in the maternal network in response to the prototypical repetitive-rhythmic social moments. […] Adapting these findings to human maternal-infant bonding, we examined whether one function of social synchrony is to "engrave" a temporal representation of the infant in the maternal caregiving network and hypothesized that synchrony and its dyad-specific rhythms may build and amplify temporal patterns in the mother's brain".

• Conceptual Precision. With all due respect to the authors, I find the terms "maternal brain," "social brain," etc. problematic. First, there is only one brain-the notion of "different brains" invokes the outdated triune brain concept. Second, talking about "the maternal brain," "social brain," and what have you is problematic because it implies a mutually exclusive set of brain regions. Yet the regions outlined herein are all equally likely to be involved in maternal behavior, social behavior, emotion, value-based decision-making etc. Finally, this terminology causes a form of essentialism that is problematic for scientists and lay people alike. Given that a paper like this will no doubt get some attention from the press, it seems all the more problematic to refer to the neural processes that are potentiated by OT following birth and that respond to an infants' cues as "the maternal brain." If you have postpartum depression is there something wrong with your maternal brain? If you don't give birth to your child or rear them from infancy, is your maternal brain deficient?

Thank you for this important comment. We agree that the name "maternal brain" is a poor choice for the neural network that sustains parental caregiving in humans and we now term it throughout the manuscript as the "human caregiving network". In the introduction we now define the mammalian caregiving network, the subcortical network that underpins mammalian mothering, and discuss how the human caregiving network evolved to integrate several cortical networks with the subcortical mammalian-general structures into a coherent "human caregiving network" that underpins caregiving and affiliation in humans.

• Conceptual Framing: Model vs. Data. The biggest issue is the mismatch between the way in which the study is conceptually framed/interpreted and the actual data and results. The study is framed around the impacts of mother-infant synchrony effects on the infant, and the long-term consequences of those effects on the infant, whereas the data actually measured are impacts of mother-infant synchrony on the mother's own brain. If the authors prefer to keep the framing of the paper as it currently stands, then I think data that more directly assesses parenting behaviors of the parents, or temperament of their children, is needed. Do the authors have any additional data about the actual parenting behaviors of the parents, or temperament of their children, which might relate to the maternal brain activity?

We agree that the current framing requires the inclusion of mother-child relational variables, particularly mother-child synchrony, and discussion of the impact of synchrony on infant development. We now included data about the mother's actual parenting behaviors – mother-child behavioral synchrony in the home environment – and its relation to the mother's neural response to the social condition in the supplementary results and methods sections.

Please note that this mother-child synchrony variable was included in our first submission but was removed in light of the editor's suggestions. We now include it as supporting and converging evidence in the supplementary material and highlight the preliminary nature of these findings in the limitation section at the end of the Discussion. In the text we also included in the introduction discussion on mother-infant synchrony and its critical role for the development of children's social-emotional competencies across childhood and adolescence (page 3-4).

Mother-infant synchrony during social interaction:

Mothers and infants were videotaped in their home during naturalistic play and this play condition was separate from the three conditions used for the fMRI scanning (unavailable, unresponsive, social). Mother-infant synchrony in the home environment was coded using the Coding Interactive Behavior (CIB) Manual (Feldman, 1998), a global rating system for social interaction that has been used across ages, cultures, and conditions, has good psychometric properties, and has yielded over 170 scientific publications. The synchrony construct of the CIB includes the codes of dyadic reciprocity, mutual adaptation, and fluency, and these codes are averaged into a "mother-infant synchrony" construct (this coding is different from the micro-coding that was used to test the three maternal conditions and demonstrate that the "social" condition indeed exposes mothers to greater social synchrony).

We tested this mother-infant synchrony construct in relation to maternal brain response to the self-social condition under placebo. Significant positive correlations emerged between mother-infant synchrony with activity in the bilateral insula, ACC and VTA. Bayesian Pearson's correlations supported the findings in the ACC and VTA. We added full description of these analyses to the supplementary material, under results and methods section.

Infant temperamental dispositions:

While we regrettably do not have a direct measure of infant temperament (typically a self-report by the mother), to address the editor's question about infant temperament, we analyzed the "infant alone" condition, a two-minute video vignette in which the infants played alone with age-appropriate standard toys. Such setting is often used in the developmental literature to measure temperament (e.g., LabTab; Goldsmith and Rothbart, 1996) and to assess infant persistent attention, sustained exploration, and affect, which are marker of emotionality and regulation, the core features of infant temperament (Rothbart, 1981). We micro-coded on a second-by-second level two behavioral parameters of toy exploration and affect, each on a scale of 1 (low) to 5 (high): sustained attention/persistent exploration (index of regulation) and affect (index of emotionality). The attention/exploration scale (attention regulation) was coded as follows: 1 = no interest in the toy; 2 = holding toy + no gaze; 3 = mouthing; 4 = manipulation of toy + short gaze; 5 = sustained exploration with focused attention. The affect scale (emotionality) was coded as follows: 1 = very negative – crying; 2 = negative – fussing; 3 = neutral; 4 = positive; 5 = very positive.

Average score of each parameter was calculated for each infant and Pearson correlations were computed with maternal brain response. As seen in Appendix 1—table 6, we found no significant correlations.

We also computed correlations between infant temperamental codes and mother-infant synchrony and again, found no correlations.

**Author response table 1. resptable1:** 

		Mother-infant synchrony				
Infants toy exploration		Pearson's r		0.159		
		p		0.468		
		Lower 95% CI		-0.271		
		Upper 95% CI		0.536		
Infants affect		Pearson's r		0.053		
		p		0.811		
		Lower 95% C		-0.368		
		Upper 95% CI		0.455		

Infant social engagement:

In addition to these codes, we used the CIB coding scheme for assessing child social behavior (social engagement) during the naturalistic mother-child interaction. The child engagement construct is the average of the following scales: infant social initiation, infant alert, infant positive affect, and infant vocalization during the interaction. Child engagement showed a positive correlation with TP and VTA response to Self and Other social stimuli under placebo. However Bayesian analyses demonstrated weak to moderate evidence for these correlations. A detailed description of these findings is now included in the supplementary material under methods section.

• Condition Confounds. The greater reliability in brain activity between sessions for social v. non-social stimuli seems like an important confound. Can the authors be sure that greater reliability in activity between sessions for social behavior is not driven by greater similarity in stimulus-driven properties (e.g., peek-a-boo in both social conditions but heterogeneous behavior in control conditions)?

In both imagining session, under oxytocin and placebo, the participants watched the exact same videos of the three conditions. In addition, at home, before filming the three conditions, mothers were given identical instructions. Therefore, there is no reason to assume stimulus-driven properties or task demands during the social condition, as these were similar across the three conditions.

• Attrition. 12/35 (~34%) is a rather high attrition rate and raises important concerns about experimenter degrees of freedom and selective attrition. This needs to be carefully addressed and acknowledged as a limitation.

The attrition was relatively high as we needed to schedule mothers for two consecutive sessions within a two-week period at a time when they had a young infant at home, and if there were unpredictable conditions (e.g., infant illness), which prevented mothers from coming again at the allotted time-span, the participant needed to be dropped from analysis. Similarly, if there were any technical problems with one of the scans, the participant was excluded. We agree that this is a limitation, which is inherent in imaging post-partum mothers twice, and we have added it to the limitation section in the Discussion (page 33).

• Interpretation. Both Reviewers commented on this…– The authors don't sufficiently deal with the fact that OT administration specifically reduces activity within a network that 'codes for' maternal behavior during social behavior. They argue that this indicates that the mother is in an anxiolytic state, but this seems a bit speculative-wouldn't she be in an anxiolytic state regardless of the stimuli being observed? It is true that OT is thought to have an "anxiolytic" effect in rat models, insofar as it diminishes neophobia of pups amongst nulliparous females. When it does so in rats, it changes the functional connectivity within the maternal network in terms of which brain regions are involved. Yet this does not necessarily describe why human mothers would have reduced activity within the general network supporting maternal behavior under OT. Clearly there is some effect of OT, but the interpretation of what a reduced BOLD signal means is far from clear. My personal take is that we still know too little about what the BOLD signal actually indexes to make too many inferences about what an overall reduction of BOLD within a network under pharmacological manipulation actually means, psychologically.

We thank the reviewers for this point and fully agree that the associations between oxytocin administration and BOLD response are far from clear and prior studies have yielded inconsistent results. For instance, several studies show decrease in amygdala activation while others show increase in amygdala activation following the same dose of oxytocin (Domes et al., 2010, 2007).

We suggest at both the Introduction and Discussion that our findings highlight the fact that OT targets specifically the "social" condition and that these OT-mediated effects are observed as a unified network's response to social stimuli, but not to non-social stimuli. These findings are consistent with the majority of the literature, which indicates that OT administration affects *social* functions (e.g., length of social gaze, altruistic behavior in economic games, etc.) to a much greater extent than non-social cognitive or attentive processes.

We also considered the few studies that examined the effects of OT administration on parents' brain, of which most showed increase to own infant stimuli under PBO, which was attenuated under OT (Wittforth-Schardt et al., 2012; Bos et al., 2018; Riem et al., 2016)

We added on page 6-7 the following:

“Yet, the effects of OT administration on BOLD response are far from clear and the literature is mixed on whether OT increases or attenuates activations of nodes within the caregiving network (Chen et al., 2017; Grace et al., 2018; Martins et al., 2020; Wang et al., 2017; Wigton et al., 2015). […] However, since other studies showed BOLD increases under OT in fathers' brain (Li et al., 2017) and as the current consensus is that OT effects are time-, person- and context-sensitive (Bartz et al., 2011), we hypothesize that OT administration would target the social condition but the direction of its effect on social stimuli remains an open question.”

To complement these findings, in the discussion we cite a recent study by Martins et al. (2020), which showed that OT administration through both intravenous and nasal pathways decreased activity in widely-distributed regions, which parallel many of our areas (amygdala, TP, hippocampus, insula etc.) during rest and the authors attribute this decrease to the increase in peripheral oxytocin, which is also shown here.

Overall, we suggest that the effect of OT on BOLD response is only one dimension encoding the effects of OT in the brain, which has been the main outcome to date. Our results suggest additional dimensions that merit further research and here we highlight the dimension of temporal patterns, relating the temporal processing of naturalistic social stimuli. Our results show that whereas OT leveled-out the magnitude of BOLD response in the network to the social condition, it did not diminish the consistency of the response in the temporal domain, particularly in areas sensitive to temporal regularities. In the discussion, we address potential mechanisms for the effect of OT on attention to temporal consistencies of the social condition, in light of the oxytocin-dopamine link that underpins bond formation and the sensitivity of dopamine neurons to temporal contingencies. In addition, and consistent with the anxiolytic model, we raise the possibility that the role of OT during labor is to attenuate maternal attention to distinct emotional states as a mechanism of survival during birth and that the high and unusual levels of OT caused by administration may have a similar function in postpartum mothers when exposed to infant stimuli; however, we emphasize that these are hypotheses that should be tested in future research.

Finally, to further test possible mechanisms of OT function, and in line with findings linking it with increase or decrease in functional connectivity in the brain (Gangopadhyay et al., 2020; Riem et al., 2012; Wittfoth-Schardt et al., 2012), we conducted an exploratory analysis of BOLD signal correlations in areas of the caregiving network under OT and PBO to test whether OT changed connectivity patterns (see Author response table 2). However, we found no such changes in our study.

**Author response table 2. resptable2:** Paired sample t test results comparing functional connectivity under OT and PBO.

ROIs connectivity	t	Df	p	Cohen's d	Lower 95% CI	Upper 95% CI
ACC-Amygdala	-0.300	22	0.767	-0.062	-0.471	0.374
ACC-VTA	-0.990	22	0.333	-0.206	-0.617	0.209
ACC-Insula	-0.366	22	0.718	-0.076	-0.485	0.334
Amygdala-Insula	-0.159	22	0.875	-0.033	-0.442	0.376

– The rationale and interpretation of the within-subject correlation (WSC) is unclear. If oxytocin is expected to upregulate neural responses within the maternal care network, in response to social stimuli, then wouldn't one expect there to be a lower WSC across the oxytocin and placebo conditions for the social condition? My apologies if I am missing something obvious here; if so, please provide a more explicit explanation within the main text.

Thanks you for this important and helpful comment. Our findings indicate that whereas the overall level of BOLD response to the social condition is attenuated under OT across the network, the temporal dynamics of the response to the unfolding social stimuli remain stable. This indicates that the decrease in BOLD response did not reduce the activity of these regions below the noise threshold, which would have degraded the temporal consistency between runs in this condition. Thus, our findings show that the OT-driven decreases in BOLD did not eliminate the responsiveness of these regions to the moment-by-moment dynamics of the social situations. Indeed, the BOLD findings suggest that the encoding of social stimuli and their modulation by OT are not fully captured by univariate analysis on BOLD-level fluctuations and more likely reflect a combination of BOLD changes with more nuanced temporal and spatial neural representations. Thus, it is possible that OT, in accordance with the anxiolytic model, provides a gain function which reduces the magnitude of BOLD responses while maintaining the temporal engram. We now address this explicitly in the discussion (page 31):

“While the magnitude of BOLD responses towards social stimuli were diminished under OT in the maternal caregiving network, the WSC encoding the temporal pattern of response to the stimuli was preserved. […] The current results suggest a possible gain function for OT in maternal caregiving networks, which modulates the signal magnitude while preserving the encoding of temporal patterns during social stimulation.”

References

Owen SF, Tuncdemir SN, Bader PL, Tirko NN, Fishell G, Tsien RW. 2013. Oxytocin enhances hippocampal spike transmission by modulating fast-spiking interneurons. *Nature* 500:458–462. doi:10.1038/nature12330

Walum H, Young LJ. 2018. The neural mechanisms and circuitry of the pair bond. *Nat Rev Neurosci* 19:643–654. doi:10.1038/s41583-018-0072-6

Williams JR, Inselt TR, Harbaught CR, Carter CS. 1994. Oxytocin Administered Centrally Facilitates Formation of a Partner Preference in Female Prairie Voles (Microtus ochrogasfer), Journal of Neuroendocrinology.

Feldman, R. (1998). *Coding Interactive Behavior (CIB) Manual*. Unpublished Manuscript. Bar Ilan University.

Goldsmith, H. H., and Rothbart, M. K. (1996). *Prelocomotor and Locomotor Laboratory Temperament Assessment Battery, Lab-TAB; version 3.0.* Technical Manual, Department of Psychology, University of Wisconsin, Madison, WI.

Rothbart, M. K. (1981). Measurement of Temperament in Infancy. *Child Dev.* 52, 569. doi: 10.2307/1129176.

Bartz JA, Zaki J, Bolger N, Ochsner KN. 2011. Social effects of oxytocin in humans: Context and person matter. *Trends Cogn Sci* 15:301–309. doi:10.1016/j.tics.2011.05.002

Bos PA, Spencer H, Montoya ER. 2018. Oxytocin reduces neural activation in response to infant faces in nulliparous young women. *Soc Cogn Affect Neurosci*. doi:10.1093/scan/nsy080

Chen X, Gautam P, Haroon E, Rilling JK. 2017. Within vs. between-subject effects of intranasal oxytocin on the neural response to cooperative and non-cooperative social interactions. *Psychoneuroendocrinology* 78:22–30. doi:10.1016/j.psyneuen.2017.01.006

Domes G, Heinrichs M, Gläscher J, Büchel C, Braus DF, Herpertz SC. 2007. Oxytocin Attenuates Amygdala Responses to Emotional Faces Regardless of Valence. *Biol Psychiatry* 62:1187–1190. doi:10.1016/J.BIOPSYCH.2007.03.025

Domes G, Lischke A, Berger C, Grossmann A, Hauenstein K, Heinrichs M, Herpertz SC. 2010. Effects of intranasal oxytocin on emotional face processing in women. *Psychoneuroendocrinology* 35:83–93. doi:10.1016/J.PSYNEUEN.2009.06.016

Gangopadhyay P, Chawla M, Dal Monte O, Chang SWC. 2020. Prefrontal–amygdala circuits in social decision-making. *Nat Neurosci*. doi:10.1038/s41593-020-00738-9

Grace SA, Rossell SL, Heinrichs M, Kordsachia C, Labuschagne I. 2018. Oxytocin and brain activity in humans: A systematic review and coordinate-based meta-analysis of functional MRI studies. *Psychoneuroendocrinology* 96:6–24. doi:10.1016/j.psyneuen.2018.05.031

Li T, Chen X, Mascaro J, Haroon E, Rilling JK. 2017. Intranasal oxytocin, but not vasopressin, augments neural responses to toddlers in human fathers. *Horm Behav*. doi:10.1016/j.yhbeh.2017.01.006

Martins DA, Mazibuko N, Zelaya F, Vasilakopoulou S, Loveridge J, Oates A, Maltezos S, Mehta M, Wastling S, Howard M, McAlonan G, Murphy D, Williams SCR, Fotopoulou A, Schuschnig U, Paloyelis Y. 2020. Effects of route of administration on oxytocin-induced changes in regional cerebral blood flow in humans. *Nat Commun* 11:1–16. doi:10.1038/s41467-020-14845-5

Neumann ID, Slattery DA. 2016. Oxytocin in General Anxiety and Social Fear: A Translational Approach. *Biol Psychiatry* 79:213–221. doi:10.1016/j.biopsych.2015.06.004

Riem MME, Bakermans-Kranenburg MJ, van IJzendoorn MH. 2016. Intranasal administration of oxytocin modulates behavioral and amygdala responses to infant crying in females with insecure attachment representations. *Attach Hum Dev* 18:213–234. doi:10.1080/14616734.2016.1149872

Riem MME, Van Ijzendoorn MH, Tops M, Boksem MAS, Rombouts SARB, Bakermans-Kranenburg MJ. 2012. No laughing matter: Intranasal oxytocin administration changes functional brain connectivity during exposure to infant laughter. *Neuropsychopharmacology* 37:1257–1266. doi:10.1038/npp.2011.313

Saarimäki H, Gotsopoulos A, Jääskeläinen IP, Lampinen J, Vuilleumier P, Hari R, Sams M, Nummenmaa L. 2016. Discrete Neural Signatures of Basic Emotions. *Cereb Cortex*. doi:10.1093/cercor/bhv086

Shamay-Tsoory SG, Abu-Akel A. 2016. The Social Salience Hypothesis of Oxytocin. *Biol Psychiatry* 79:194–202. doi:10.1016/j.biopsych.2015.07.020

Ulmer-Yaniv A, Salomon R, Waidergoren S, Shimon-Raz O, Djalovski A, Feldman R. 2020. Synchronous Caregiving from Birth to Adulthood Tunes Humans’ Social Brain. *bioRxiv Prepr*.

Wang D, Yan X, Li M, Ma Y. 2017. Neural substrates underlying the effects of oxytocin: A quantitative meta-analysis of pharmaco-imaging studies. *Soc Cogn Affect Neurosci* 12:1565–1573. doi:10.1093/scan/nsx085

Wigton R, Radua J, Allen P, Averbeck B, Meyer-Lindenberg A, McGuire P, Sukhi S, Fusar-Poli P. 2015. Neurophysiological effects of acute oxytocin administration: Systematic review and meta-analysis of placebo-controlled imaging studies. *J Psychiatry Neurosci* 40:E1–E22. doi:10.1503/jpn.130289

Wittfoth-Schardt D, Gründing J, Wittfoth M, Lanfermann H, Heinrichs M, Domes G, Buchheim A, Gündel H, Waller C. 2012. Oxytocin modulates neural reactivity to children’s faces as a function of social salience. *Neuropsychopharmacology*. doi:10.1038/npp.2012.47

Saarimäki H, Gotsopoulos A, Jääskeläinen IP, Lampinen J, Vuilleumier P, Hari R, Sams M, Nummenmaa L. 2016. Discrete Neural Signatures of Basic Emotions. *Cereb Cortex*. doi:10.1093/cercor/bhv086

Ulmer-Yaniv A, Salomon R, Waidergoren S, Shimon-Raz O, Djalovski A, Feldman R. 2020. Synchronous Caregiving from Birth to Adulthood Tunes Humans’ Social Brain. *bioRxiv Prepr*.

[Editors' note: further revisions were suggested prior to acceptance, as described below.]

Essential revisions:Both Reviewers remained concerned about the conceptual framing and underscored the need to temper the claims to better align with the approach and results (i.e. avoid "overselling")…• Remove the temporal engram concept from the manuscript entirely.– The only thing I still have concerns about is the temporal engram argument. It is not clear to me that the authors have demonstrated the existence of a "temporal engram" for the infant – at least in their definition – using their WSC analyses.– As I understand their logic, the point of a temporal engram is that it ""engrave(s)" a temporal representation of the infant in the maternal caregiving network and hypothesized that synchrony and its dyad-specific rhythms may build and amplify temporal patterns in the mother's brain."– However, the authors' findings do not support this argument – they show WSC for the social condition but no differentiation by self v. other, meaning that the network thought to be involved in representing social stimuli shows more reliable activity to social stimuli than non-social stimuli across instances, which is not in and of itself surprising.– Likewise, they do not show specific activity for the "partners presence" (i.e., the infant).– I am also still not convinced that the greater reliability is not a confound of stimulus properties. I understand that they saw the same stimulus at time 1 and time 2, but this does not solve the problem that there could be more randomness in neural responses to videos of unresponsive mothers because these videos are more boring, less salient, and generally entrain neural processes to a lesser degree.– I'm certainly open to the interpretation that greater WSC to social v non-social stimuli is meaningful in some important way – but I think the authors are overselling it by trying to infer that this is a 'temporal engram.' My recommendation would be that they remove the temporal engram concept from the manuscript entirely.

Thank you for this important point. The concept of "temporal engram" is now removed from the manuscript and we altered the abstract, introduction, and discussion accordingly. The WSC results are now described more operatively, we emphasize time and again that these analyses were exploratory and should be treated with extreme caution, and indicate that the results require further research and validation and suggest areas requiring further work (pages 26-27).

In addition, all the points mentioned by the reviewers above (salience related to social versus non-social stimuli, no control for partners presence, and the need to provide more control conditions are now brought up in the Discussion as limitations of these analyses, and the findings are described as preliminary).

• I think the framing of the paper needs to be altered so that the results are not misinterpreted; this would include changes in the abstract, introduction and Discussion sections to be more explicit the study's findings and being careful to not go beyond the data.

Thank you for this important point. We have made significant changes in the manuscript to tone down the statement and stay close to the findings. Specifically, we have made changes in the abstract (page 2 lines 7-9), introduction (page 5 lines 6-7; 19-21, page 6 lines 8-9, page 7 lines 6-8, page 7 lines 23-25, page 8 lines 1-5) and discussion (page 26 lines 23-25, page 27 lines 1-9, page 27 lines 18-20, page 28 lines 19-25, page 29 lines 1-3). We made the introduction more concise and focused on our hypotheses, repeatedly emphasize that the WSC analyses were exploratory, and kept the discussion focused on the specific findings.

– There appears to be some modest support for the larger framework of the paper, with the mother's brain responses within particular nodes of the human caregiver network exhibiting modest relations with infant social engagement. Similarly, the mother's brain responses within particular nodes also exhibited some relations to mother-infant synchrony.– Yet the more crucial analyses were not significant: no brain regions within the human caregiving network exhibited activity that was predictive of infant temperament, and moreover, mother-infant synchrony was not related to the infant's temperament.

We added discussion on the findings related to infant social engagement and the absence of findings in relation to infant temperament.

However, we wish to respond to the reviewer's point that "the more crucial analysis were non-significant since (a) no region in mother's caregiving network predicted infant temperament and (b) synchrony was not related to temperament. Infant temperament was never a component of our study.

Temperament was never a part of the study and we did not hypothesize that infant temperament would be related to activations of the caregiving network in postpartum mothers. Developmental theories on attachment and temperament (e.g., Bowlby on attachment and Sroufe, Rothbart, and Bates on temperament) consider the two dimensions, attachment and temperament, to be clearly distinct. Sensitive mothers are expected to form attachment with infants of any temperament (albeit some infants may pose greater challenges) and no study, to our knowledge, demonstrated associations between reorganization of the maternal brain and infant temperament. Activations of the mother's caregiving network has been linked with maternal factors, such as anxiety, depression, representations of own caregiving, and circling hormones such as oxytocin and cortisol and we cite these findings in the introduction (second paragraph).

Our key hypothesis was that mother-infant synchrony is a crucial early social experience that is associated with reorganization of the mother's brain. We cite extant research to support this hypothesis. The only reason we included measures of infant temperament was in response to the reviewers' requests in the previous round to examine whether neural activations were correlated with any measure of infant temperament. Since we did not have standard self-report measures of temperament (we did formulate hypotheses related to temperament), we coded temperament-related variables from a video clip of the infant alone. This is not a standard way to measure temperament but to address the reviewer's point we now included the temperament results, which were previously reported only in the letter, in the SM.

With regards to the associations of temperament and synchrony, such associations were not part of the current study's hypotheses, the temperament measure is not standard, and we therefore did not compute correlations between these "temperament" measures and synchrony. We now cite this as study limitation (page 30).

In contrast, the "child social engagement", which can be used as a proxy for the infant's temperamental "sociality", correlated with activations of the maternal caregiving network in both the VTA (r_p_=0.438, p=0.037) and TP (r_p_=0.521, p=0.011). This variable was, again, not a focus of our study, and was included in response to the reviewers' request.

[Editors' note: further revisions were suggested prior to acceptance, as described below.]

Essential revisions:Both Reviewers remain moderately concerned about the conceptual framing. They re-emphasized the need to temper the claims to better align with the approach and results (i.e. avoid "overselling") and to sketch out the most important challenges for the future• A reviewer notes, My primary concern with this article remains that the authors appear to go beyond the data in the overall framing of the paper, specifically, their claims about "cross-generational transmission". In particular, the last line of their most recent abstract reads: "Findings describe how mother's brain compiles and amplifies these precious social moments to generate dyad-specific patterns that initiate the cross-generational transmission of human sociality." Additionally, the last two sentences of the first paragraph of the Discussion section: "Across mammalian species, the cross-generation transmission of sociality is initiated by the consolidation of the maternal neural network from which, through caregiving behavior, a similar network is sculpted in the infant's brain (Feldman, 2020; Numan, 2020). Our study uniquely tests the response of the human caregiving network in the maternal brain to these social moments, versus other moments of non-social mother-infant presence, to shed further light on the cross-generation transmission of human sociality". I emphasized this issue in the most recent round of review. However, the authors unfortunately seem to have missed the central point of my comments.• Based on a discussion with the Reviewers, we collectively recommend the following changes to the manuscript:1. Eliminate the phrase "cross-generational transmission" (and similar) from the Abstract2. Modify the Discussion, revising the sentence in question to read something like: "Our study uniquely tests the response of the human caregiving network in the maternal brain to these social moments, versus other moments of non-social mother-infant presence, to shed further light on how human mother brains may change following birth and vary with the caregiving experience with one's infant."3. Provide a brief, but explicit discussion of ways in which future work could rigorously address "cross-generation transmission".

In the final version, we made the three changes you requested.